# A threshold level of JNK activates damage-responsive enhancers via JAK/STAT to promote tissue regeneration

John W. Quinn, Mariah C. Lee, Chloe Van Hazel, Melissa A. Wilson and Robin E. Harris*

## ABSTRACT

Tissue regeneration requires precise activation and coordination of genes, many of which are reused from development. Although key factors have been identified, how their expression is initiated and spatially regulated after injury remains unclear. The stress-activated MAP kinase JNK is a conserved driver of regeneration and promotes expression of genes involved in proliferation, growth and cell fate changes in *Drosophila*. However, how JNK selectively activates its targets in damaged tissue is not well understood. We have previously identified damage-responsive, maturity-silenced (DRMS) enhancers as JNK-activated elements that are crucial for regeneration. Here, we show that cell death is dispensable for the activation of these enhancers, which only depend on JNK and its immediate downstream effectors. One of these is JAK/STAT, which acts as a direct, additional input necessary to expand enhancer activity into the wound periphery where JNK alone is insufficient. Furthermore, we demonstrate that a threshold level of JNK is required to initiate enhancer activation. Together, our findings reveal how JNK and JAK/STAT signaling cooperate to drive spatially and temporally regulated gene expression through damage-responsive enhancers, ensuring proper regenerative outcomes.

KEY WORDS: *Drosophila*, Regeneration, Enhancer, Gene regulation, JNK, JAK/STAT

## INTRODUCTION

Regeneration, the repair and regrowth of damaged tissues, occurs across numerous species in diverse tissue contexts, though often poorly in humans (Sanchez Alvarado and Tsonis, 2006; Li et al., 2015; Tanaka and Reddien, 2011). It has been well established that regenerative growth often reuses developmental signaling; for example, FGF8, retinoic acid and Sonic Hedgehog are crucial for both salamander limb development and for its regrowth following amputation, with striking similarities in the temporal and spatial expression patterns of each factor (Tanaka, 2016). Thus, regenerative failure likely stems not from an absence of necessary genetic factors, but the inability to reinitiate their expression in a coordinated fashion after injury. Until recently it was unclear how a single gene could be expressed differently in the contexts of

regeneration and development, but the discovery of damage-responsive (DR) enhancers have provided an explanation (Rodriguez and Kang, 2020; Harris, 2022; Suzuki and Ochi, 2020). These enhancers have been identified in diverse organisms and tissues, and often share traits like modularity, common signaling inputs and epigenetic regulation (Rodriguez and Kang, 2020; Harris, 2022; Suzuki and Ochi, 2020). Their study has already revealed key insights into the identity, initiation and behavior of regeneration-promoting genes, and enabled strategies to improve regeneration even in non-regenerative organisms (Harris et al., 2016; Kang et al., 2016; Yan et al., 2023). Advancing our understanding of these elements is therefore essential.

To study how regenerative gene expression is initiated and regulated, we use the *Drosophila* wing imaginal disc, a larval epithelial tissue that forms adult wing structures, as a model (Tripathi and Irvine, 2022). Extensively used to investigate fundamental aspects of development (Tripathi and Irvine, 2022; Beira and Paro, 2016), the disc has also become a powerful platform for dissecting the genetics underlying regeneration (Worley et al., 2012; Worley and Hariharan, 2022; Fox et al., 2020). Our previous work using the wing disc identified discrete genomic regions that change in accessibility after damage, correlating with altered expression of nearby genes, suggesting they likely function as regeneration-associated enhancers (Harris et al., 2016, 2020). Using a genetic ablation system that we developed called DUAL Control (DC) (Harris et al., 2020; Harris, 2023), we focused on two regeneration-promoting genes, *wingless* (*wg*, the *Drosophila* ortholog of vertebrate *Wnt1*) and *Matrix metalloproteinase 1* (*Mmp1*). Both are strongly induced by damage in regeneration-competent early third instar (L3) discs, but only weakly induced in late L3 discs that fail to regenerate (Harris et al., 2016, 2020). We identified enhancers at both loci with similar modular structures: a DR region activated by injury, and a maturity-silencing (MS) region that limits enhancer activity in older discs via epigenetic silencing. Together, these modules form DRMS enhancers that drive damage-induced gene expression, which subsequently becomes limited in late L3 discs, even in the presence of activating signals (Harris et al., 2016). Importantly, their modularity allows independent study of the DR and MS regions, a fact we have taken advantage of here. DRMS enhancers are activated by JNK signaling directly via AP-1 sites (Harris et al., 2016). Using their altered epigenetic signature, we subsequently identified dozens of additional putative DRMS enhancers with similar features, suggesting coordinated regulation of multiple genes by this mechanism (Harris et al., 2020). JNK activation and epigenetic regulation appear to be core aspects of regeneration in wing discs (Vizcaya-Molina et al., 2018), and for tissue repair via damage-responsive enhancers in other species (Rodriguez and Kang, 2020; Harris, 2022). Thus, understanding these enhancers has broader implications for regenerative biology.

Crucially, JNK signaling during normal development, such as in the embryo or wing disc notum, fails to activate these enhancers or

School of Life Sciences, Arizona State University, 427 E Tyler Mall LSE 229, Tempe, AZ 85287-4501, USA.

*Author for correspondence (robin.harris@asu.edu)

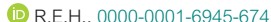 R.E.H., 0000-0001-6945-6741

their regenerative targets (Harris et al., 2016). This suggests that either an injury-specific environment allows JNK to induce enhancer activity, or that additional pathways are required, or both. To explore this, we have investigated the involvement of additional regulatory factors. We find that, while JNK is necessary for DR enhancer activation, cell death and its resulting downstream signaling events are not. Moreover, JAK/STAT, a target of JNK, is required for full enhancer activation and proper spatial expression. Finally, by isolating JNK from the feedforward mechanisms that arise following cell death, we show that distinct JNK thresholds control target gene activation: high JNK levels at the center of a wound are sufficient to activate DR enhancers alone, while lower levels in the wound periphery require the additional input of JAK/STAT. At the lowest levels of JNK, where JAK/STAT is not induced, only a limited subset of targets respond, such as those expressed during development like *puckered* (*puc*). Thus, DR enhancers integrate JNK intensities generated by a wound alongside JAK/STAT input to direct gene expression with spatial and temporal precision during regeneration.

## RESULTS

### Cell death is dispensable for activation of the regeneration program via DR enhancers

To investigate how DR enhancers are activated during regeneration, we examined an enhancer at the WNT locus known to drive *wg* and *Wnt6* expression upon injury (DRMS$^{WNT}$; Harris et al., 2016, 2020; Gracia-Latorre et al., 2022) (Fig. 1A). We used a GFP reporter driven by the DR module of the enhancer (*DR$^{WNT}$-GFP*) (Fig. 1A), which is inactive in undamaged tissue and unaffected by developmental stage when isolated from the MS region (Harris et al., 2016) (Fig. 1B). We induced damage using the DC ablation system (Harris et al., 2020; Harris, 2023), which triggers apoptosis in the distal pouch via heat-shock-induced expression of activated *hemipterous* (DC$^{hepCA}$) (Fig. 1C). Cell death activates the regeneration program, observable at ∼6 h after heat shock (AHS) until ∼48 h AHS, with significant regenerative processes represented at the 18 h time point (Harris, 2023) (Fig. 1D). Unlike other ablation systems that express a cell death stimulus continuously over an extended 20-40 h period (Smith-Bolton et al., 2009; Bergantinos et al., 2010; Santabarbara-Ruiz et al., 2015), the DC system generates an acute injury that separates ablation from regenerative processes, making it ideal for dissecting DR enhancer activation. The DC system also includes a heat-shock-activated pouch GAL4 driver (*hs-FLP; DVE≫GAL4*) to manipulate gene expression in the surrounding regenerating cells (Harris, 2023) (Fig. 1C,D).

*DR$^{WNT}$-GFP* is activated by JNK signaling (Harris et al., 2016), likely directly via AP-1 sites, as their removal blocks its activity (Harris et al., 2016). To better characterize this, we monitored *DR$^{WNT}$-GFP* alongside a JNK reporter (*AP-1-RFP*) from 0 h to 36 h AHS (Fig. 1E). GFP and RFP were first co-expressed at ∼6 h AHS and largely overlapped thereafter (Fig. 1E, 6 h, arrowhead). By 12 h AHS, GFP and/or RFP-expressing cells that were Dcp-1-positive were observed, representing dying cells undergoing clearance, while Dcp-1-negative cells indicate those activating the regeneration program (Fig. 1F,F′ and Fig. S1A-C, arrowheads). This separation of dying and regenerating cells was also observed using a *lexAop-RFP* transgene to label cells undergoing ablation (Fig. S1D). At 18 h-36 h, while a significant overlap of GFP and RFP existed (Fig. 1E; Fig. S1B), many cells with strong DR$^{WNT}$ enhancer activity showed only weak JNK activity (Fig. S1B, arrowheads). A nuclear localized DR$^{WNT}$ reporter (*DR$^{WNT}$[NLS]*) confirmed this, with strong enhancer activity in the wound periphery despite low JNK (Fig. 1F-F‴′, arrowheads in F‴,F‴′). Quantification demonstrated a

separation of high JNK in the wound center and lower JNK in the periphery (Fig. 1F‴′,G). Although we cannot rule out a difference in fluorophore perdurance of these reporters, the expression of the known JNK target *Mmp1* mirrors this pattern (Fig. S1E), suggesting that JNK alone may not fully explain DR$^{WNT}$ enhancer activity. As DR$^{WNT}$ represents only the damage-responsive portion of the full DRMS$^{WNT}$ enhancer, we also examined a reporter for the entire DRMS region (*DRMS$^{WNT}$-GFP*, Fig. S1F). It showed the same colocalization with JNK-positive cells, with the only difference being weaker GFP signal due to the basal silencing by the MS region (Harris et al., 2016). Thus, the full length DRMS enhancer behaves similarly to the DR module in its overlap with JNK.

Importantly, developmental JNK signaling (e.g. acting during dorsal closure in the embryo or thorax closure of notum (Zeitlinger and Bohmann, 1999; Agnes et al., 1999; Harden, 2002; Kiehart et al., 2017) failed to activate the DR$^{WNT}$ enhancer, even where JNK reporters and targets like *puc* are active (Fig. S1G,H). Other JNK-responsive genes like *Mmp1* (Uhlirova and Bohmann, 2006), *Ilp8* (Katsuyama et al., 2015) and JAK/STAT targets (La Fortezza et al., 2016) were also not expressed (Fig. S1I). A number of these factors are associated with DRMS enhancers (Harris et al., 2020). These findings imply that additional damage-induced factors are required, or that JNK alone is sufficient but the level of JNK activity dictates the activation of developmental versus damage-specific targets.

To test if JNK alone is sufficient to activate DR$^{WNT}$, we induced JNK signaling while blocking apoptosis to avoid downstream effects of cell death. Apoptotic cells can release extracellular mitogens like Wg and Dpp (Fogarty and Bergmann, 2017; Morata et al., 2011), and generate reactive oxygen species (ROS), triggering JNK in neighboring cells and apoptosis-induced proliferation (AiP) (Fogarty et al., 2016; Amcheslavsky et al., 2018; Pinal et al., 2018; Esteban-Collado et al., 2024; Diwanji and Bergmann, 2017). To minimize these effects, we used DC$^{hepCA}$ to activate JNK and co-expressed *UAS-mir(RHG)* (Siegrist et al., 2010) to inhibit *rpr*, *hid* and *grim*, blocking apoptosis downstream of AP-1 but upstream of caspase activation. This prevents the signaling events associated with AiP, 'undead' cells and feedforward JNK signaling (Fogarty and Bergmann, 2017; Su, 2015; Martín et al., 2009), including the generation of ROS via Dronc/Duox (Fogarty et al., 2016; Amcheslavsky et al., 2018; Khan et al., 2017), which was significantly reduced although not eliminated in this background (Fig. S6D). Under these conditions, *DR$^{WNT}$-GFP* can still be activated (Fig. 1H), as well as the enhancer's downstream target *wg* (Fig. 1H, arrowhead) and other targets regulated by DR enhancers like *Mmp1* (Fig. 1I). This shows that JNK and its immediate downstream effectors are sufficient to trigger the regeneration program via DR enhancers, while cell death and associated signaling events are not required. Interestingly, although blocking cell death in this way prevented undead cell formation and AiP, some ectopic growth still occurred, resulting in larger overall pouch tissue (Fig. S1J,K). We speculate that this could be due to the initial activation of growth-promoting factors like Wg (Fig. 1H), while the inability to sustain feedforward JNK signaling may limit this growth and prevent the formation of neoplastic tumors observed when apoptosis is blocked at the level of caspase activity (Martín et al., 2009; Perez-Garijo et al., 2004; Ryoo et al., 2004).

Together, these findings show that JNK can activate DR$^{WNT}$ independently of apoptosis. Moreover, the incomplete overlap between JNK and DR$^{WNT}$ activity, and their distinct developmental expression patterns, suggest additional inputs may modulate enhancer activation, though they likely lie outside cell-death induced signaling events.

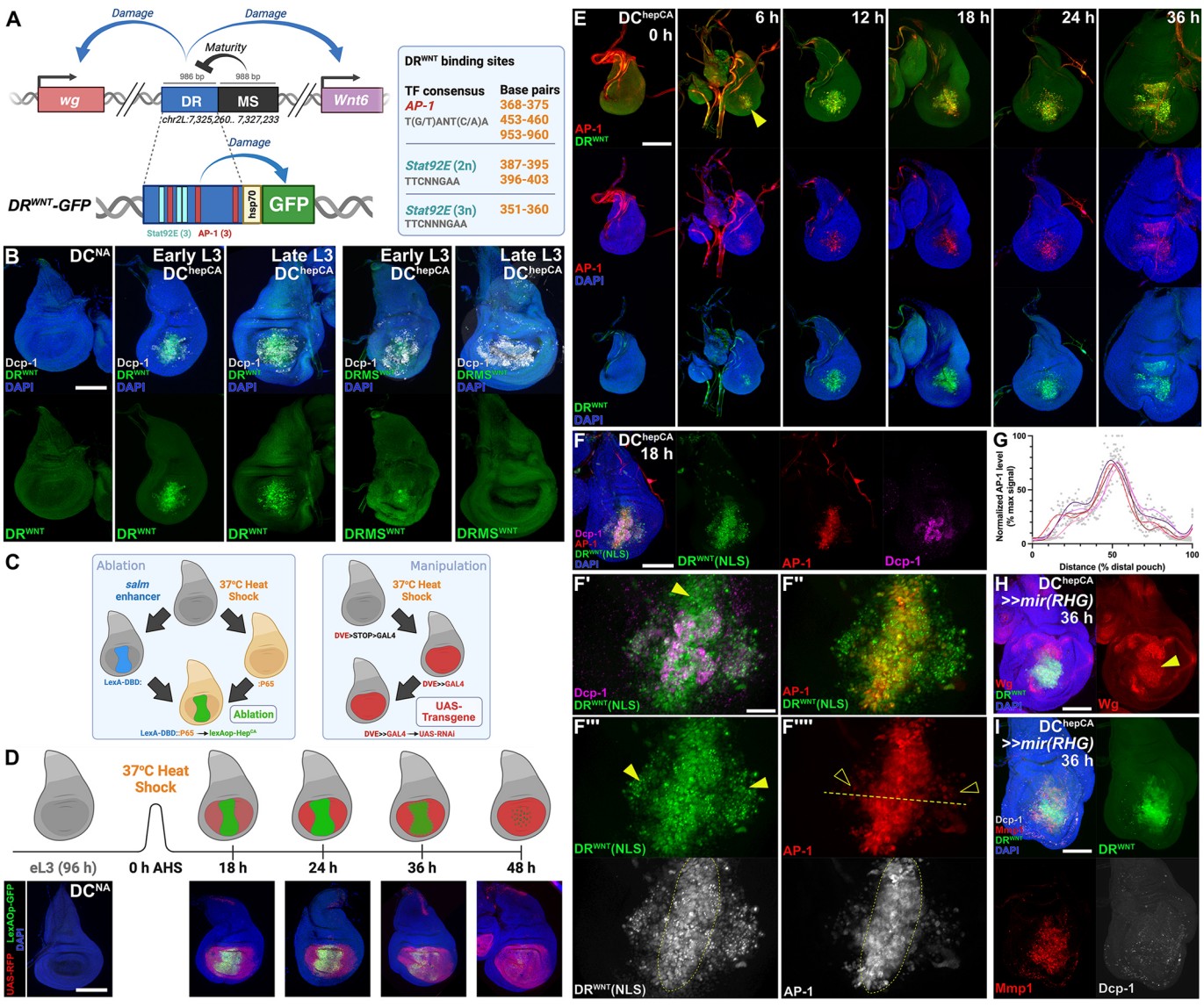

**Fig. 1. DR^WNT and the regeneration program can be activated independently of cell death.** (A) Schematic of the DRMS^WNT regulatory element (top) with separable damage-responsive (DR) and maturity-silencing (MS) regions controlling *wg* and *Wnt6* expression. Created in BioRender. Harris, R. (2025) https://BioRender.com/i6glwid. AP-1 and Stat92E binding sites in the DR region are noted (right). The DR^WNT reporter (bottom) comprises the enhancer driving GFP via an *hsp70* promoter. (B) DR^WNT and DRMS^WNT reporters in discs ablated with DC^hepCA at early (84 h AED) or late (108 h AED) L3. DR^WNT is activated in both stages, while DRMS^WNT is only active in early L3. Activation is not seen in non-ablating controls (DC^NA). (C) The DUAL Control (DC) ablation system uses heat shock to activate LexA-based ablation in the distal pouch (green) and GAL4-driven gene manipulation across the pouch (red) (Harris, 2023). Created in BioRender. Harris, R. (2025) https://BioRender.com/2djxciu. (D) Schematic of ablation timing with corresponding discs. *LexAop-GFP* (green) marks ablation domain; *UAS-RFP* (red) marks GAL4 activity. Created in BioRender. Harris, R. (2025) https://BioRender.com/2djxciu. Ablation is transient, while GAL4 expression persists through regeneration. A non-ablating version of DC (DC^NA) is used to show reporter dynamics. (E) Time course of *DR^WNT-GFP* (green) and *AP-1-RFP* (red) reporters in DC^hepCA-ablated discs shows overlapping activation from 6 to 36 h AHS. (F) Disc with DR^WNT[NLS]-GFP (green), AP-1-GFP (red), and Dcp-1 (magenta) at 18 h AHS. (F′-F‴) DR^WNT activity occurs in cells lacking Dcp-1 (F′, yellow arrowhead). DR^WNT (F‴, green) is active at the wound periphery, where AP-1 (F⁗) is reduced. Dotted circle in F‴ and F⁗ indicates the observed separation of high and low JNK activity. Line in F⁗ shows plane of quantification in G. Yellow arrowheads in F‴ and corresponding open arrowheads in F⁗ indicate cells with high DR^WNT activity (green) but low AP-1 activity (red). (G) Quantification of AP-1 fluorescence profiles across wounded discs (*n*=5 discs), normalized to background fluorescence of the disc and shown as a percent of the maximum signal detected. Individual measurement points are shown (light gray), splines created by LOWESS analysis using 10 points per window. (H,I) Discs bearing the DR^WNT reporter (green) ablated by DC^hepCA and expressing *UAS-mir(RHG)* to block cell death, imaged at 36 h AHS. (H) JNK in the absence of cell death activates Wg (red), overlapping the DR^WNT reporter (green). (I) Mmp1 (red) is induced alongside DR^WNT in the absence of Dcp-1 (white), confirming activation is cell-death-independent. Scale bars: 50 μm (B,D,E,F,H,I); 15 μm (F-F⁗).

## Transcriptomic analysis suggests JAK/STAT can regulate DR enhancers

JNK activates many pathways during regeneration (Pinal et al., 2019), many of which could contribute to DR enhancer regulation. To identify potential candidate pathways, we used next-generation sequencing approaches. Previously, we performed ATAC-seq on damaged and undamaged early and late L3 discs to identify genomic regions with altered chromatin accessibility, thus potentially representing DR or DRMS enhancers (Fig. 2A) (Harris et al., 2020). This analysis revealed 243 unique regions with increased damage-induced accessibility; 222 regions in early L3 and 33 in late L3 (Harris et al., 2020). We re-analyzed these regions for enriched transcription

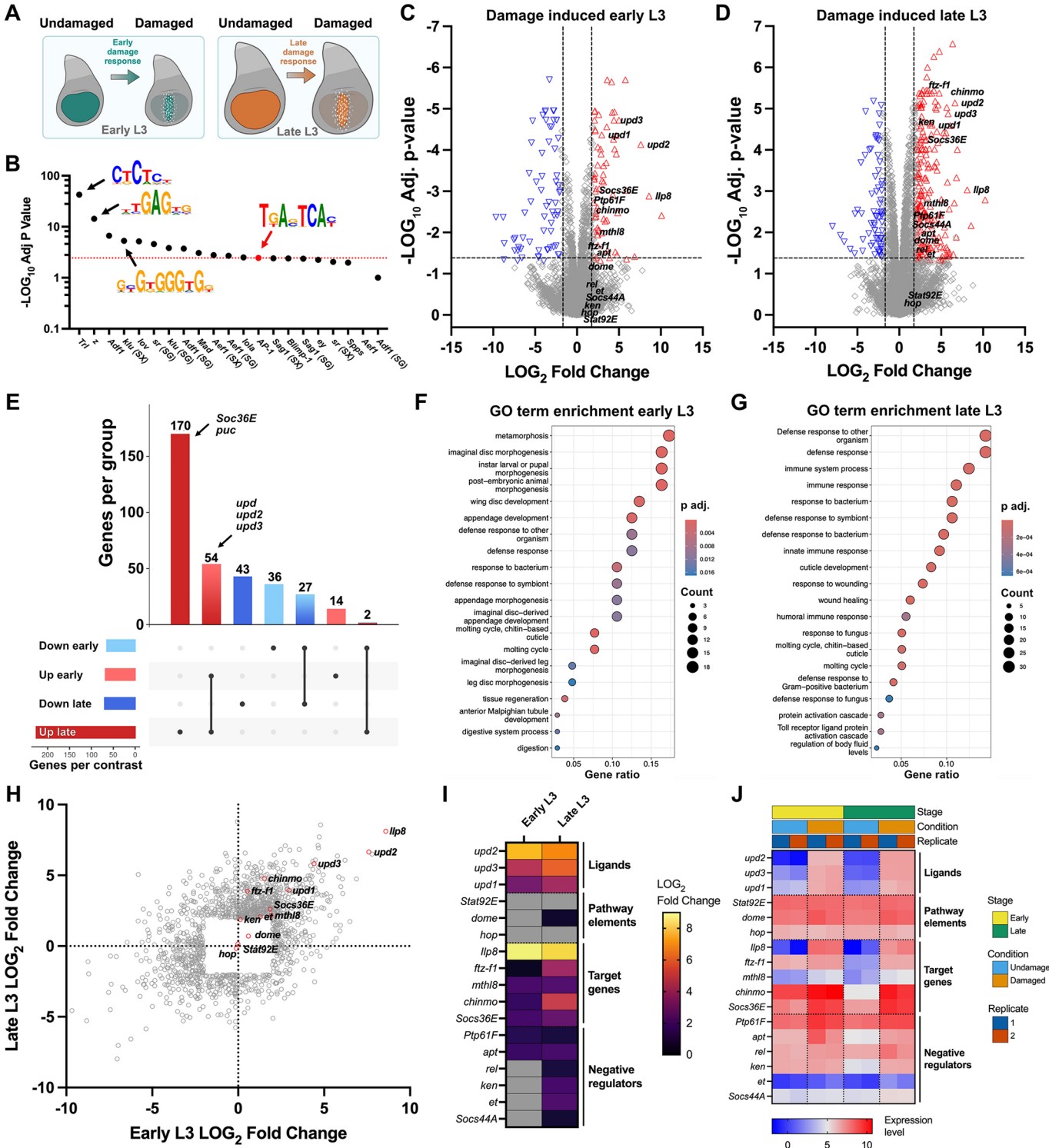

**Fig. 2. Transcriptomics and motif enrichment identify JAK/STAT as a putative regulator of DR enhancers.** (A) Schematic of RNA-seq and ATAC-seq (Harris et al., 2020) conditions used to identify damage-induced changes in early and late third instar (L3) wing discs. Discs were ablated using *rn^ts>egr* (*rn-GAL4, GAL80ts, UAS-egr*) for 40 h following culture at 18°C for 168 h (early) or 216 h (late) after egg deposition (AED), and processed immediately after ablation. (B) Motif enrichment analysis (AME) of 222 DR peaks identified in early L3 discs using 7456 static peaks as background. Enriched motifs include binding sites for chromatin regulators *Trl*, *z*, *klu* and JNK pathway factors *kay/jra* (AP-1). (C,D) Volcano plots showing differentially expressed genes (DEGs) in response to damage in early (C) and late (D) L3 discs. Genes are plotted by log₂ fold change (LOG₂FC) and −log10 adjusted *P*-value. JAK/STAT pathway-related genes are highlighted. (E) Upset plot showing overlap of DEGs between early and late L3 stages and damage conditions. Horizontal bars indicate the number of DEGs in each group; vertical bars show shared genes across conditions. *upd* ligands are upregulated in both early and late damaged discs, while *puc* and *Socs36E*, negative regulators of JNK and JAK/STAT, are upregulated only in late L3. (F,G) Gene ontology (GO) enrichment among DEGs in early (F) and late (G) L3 damaged discs. The *x*-axis shows gene ratio; dot size reflects the number of genes in each category; color indicates adjusted *P*-value. (H) Comparison of LOG₂FC values between early and late L3 stages for all genes. JAK/STAT pathway genes are labeled in red, others in gray. (I) Heatmap showing damage-induced LOG₂FC in JAK/STAT pathway components, including ligands, signal transducers, targets and negative regulators. Gray indicates non-significant changes (*P*>0.05). (J) Heatmap of JAK/STAT pathway gene expression across developmental stages (early versus late), damage conditions (undamaged versus damaged) and biological replicates. Red indicates higher expression, blue indicates lower.

factor (TF) binding motifs that might indicate regulatory signaling pathways (see Materials and Methods). Motif analysis of only early L3 DR regions (Fig. 2B), and of all DR regions in early and late L3 discs combined (Fig. S2A), revealed enrichment for the chromatin modifier *Trithorax-like* (*Trl*), the DNA binding protein *zeste* (*z*), which is involved in chromatin-targeted regulation of gene expression, and two early growth response (EGR) family factors, *klumpfuss* (*klu*) and *stripe* (*sr*). The enrichment of these two factors is notable, as previous work in the highly regenerative acoel worm *Hofstenia miamia* found that Egr functions as a pioneer factor to directly regulate early wound-induced genes and activate the entire gene regulatory network necessary for whole body regeneration (Gehrke et al., 2019). AP-1 (*kay/jra*) motifs were also enriched (Fig. 2B; Fig. S2A), but at a lower significance level than expected for a known regulator of DR enhancers ($P_{adj}$=3.59×10$^{-3}$ in early L3 DR regions, $P_{adj}$=1.82×10$^{-5}$ in all DR regions), suggesting this analysis can identify factors regulating DR enhancer chromatin accessibility, but may not be sensitive enough to identify TFs controlling their activity.

To complement this approach, we performed RNA-sequencing (RNA-seq) on discs under the same conditions as the ATAC-seq experiment (Fig. 2A) to identify genes that are differentially expressed upon damage. In early L3 discs 569 genes were altered (338 upregulated, 231 downregulated, $P$<0.05, Log$_2$FC>0.4; Fig. 2C; Table S1), while late L3 discs had 1356 differentially expressed genes (821 upregulated and 535 downregulated; Fig. 2D; Table S1). Gene ontology analysis identified enrichment for transcriptional changes related to imaginal disc morphogenesis, development and tissue regeneration in early L3 discs, and immunity, wound healing and defense responses in late L3 discs (Fig. 2F,G). Comparing early and late L3 regenerating discs identifies both stage-specific and shared damage-induced genes, (Fig. 2E; Table S2), including all three upd ligands of the JAK/STAT pathway upregulated at both stages (Fig. 2E). JAK/STAT signaling is known to function during stress responses (Katsuyama et al., 2015; La Fortezza et al., 2016; Herrera and Bach, 2019; Verghese and Su, 2016; Ahmed-de-Prado et al., 2018; Jaiswal et al., 2023; Floc'hlay et al., 2023), being activated downstream of JNK in these contexts (Santabarbara-Ruiz et al., 2015; Katsuyama et al., 2015; La Fortezza et al., 2016; Ahmed-de-Prado et al., 2018; Pastor-Pareja et al., 2008; Worley et al., 2018). While the expression of the JAK/STAT core pathway components, *domeless* (*dome*), *hopscotch* (*hop*) and *Stat92E* remain largely unchanged in our RNA-seq data (Fig. 2H-J), several known targets such as *Ilp8* (Katsuyama et al., 2015), *ftz-F1* (Wang et al., 2013), *mthl8* (Mukherjee et al., 2006), *chinmo* (Flaherty et al., 2010) and *Socs36E* (Karsten et al., 2002) are upregulated by damage in both early and late L3 discs, suggesting pathway activation and thus the potential to regulate DR enhancers. This specific transcriptomic behavior of ligands, pathway components and target genes mirrors that of the known DR enhancer regulator, JNK (Fig. S2B-E). Furthermore, although the binding motif for Stat92E (Yan et al., 1996) is not globally enriched in the DR regions versus static peaks identified by ATAC-seq (Fig. 2B; Fig. S2A), a targeted search found that 50.62% (123/243) of DR regions contain Stat92E binding sites, compared to 68.72% (167/243) with AP-1 sites (FIMO tool $P$<0.001, meme-suite.org). This aligns with published single cell ATAC-seq data showing AP-1 and Stat92E motif enrichment in chromatin regions activated in wound responsive cells (Floc'hlay et al., 2023). Notably, this enrichment occurs within a specific subset of cells within the wound, which may explain its absence from the analyses of our data that used whole discs (Harris et al., 2020). Together, these findings suggest that JAK/STAT signaling may act alongside JNK to regulate DR enhancers.

Interestingly, despite damage-induced *10×STAT-GFP* reporter activity being diminished in ablated late L3 discs (Harris et al., 2020), our RNA-seq data showed robust upd ligand expression. However, it also shows upregulation of several negative regulators of JAK/STAT (Fig. 2I,J), such as *apontic* (*apt*), which is known to restrict pathway activity (Harris et al., 2020; Starz-Gaiano et al., 2008). This suggests that while JAK/STAT signaling is still induced in late L3, its output might be limited by stage-specific negative regulators.

## JAK/STAT signaling coincides with DR enhancer activity

To test whether JAK/STAT regulates DR enhancers, we examined the spatial and temporal dynamics of JAK/STAT and JNK signaling using a fast-turnover GFP reporter for JAK/STAT (*10×STAT-DGFP*) (Bach et al., 2007) and *AP-1-RFP* as before (Chatterjee and Bohmann, 2012). Before injury, JAK/STAT was active in the hinge, whereas JNK was absent (Fig. 3A). Following DC$^{hepCA}$ ablation, JNK was detected by ~6 h and coincided with DR$^{WNT}$ (Fig. 1E). However, JAK/STAT was not detected in the pouch until 12 h AHS (Fig. S3A), consistent with its activation downstream of JNK. By 18 h AHS, JNK could be distinguished in regions of higher (distal) and lower (proximal) activity (Fig. 3B,D; Fig. S3A), while JAK/STAT appeared across both regions (Fig. 3B). At 24 h AHS, JNK levels remained elevated distally (Fig. 3C; Fig. S3A), but JAK/STAT activity declined particularly in the distal pouch (Fig. 3C,D; Fig. S3A). Quantification shows that these patterns diverged over time (Fig. 3F,G), with JNK remaining centrally within the wound, and JAK/STAT localizing more peripherally, until both pathways declined by 36 h AHS (Fig. 3F,G; Fig. S3A,B). These findings agree with recently published work that used a different genetic ablation system (*rn-GAL4/GAL80ts/UAS-egr*) to demonstrate that JNK and JAK/STAT are activated in response to injury and subsequently resolve into separate regions due to a mutually repressive interaction (Jaiswal et al., 2023). This separation allows JNK-positive cells to transiently pause proliferation, while JAK/STAT promotes it in surrounding cells (Jaiswal et al., 2023). This separation is also observed via single cell analyses (Floc'hlay et al., 2023). Using DC$^{hepCA}$, we observed a similar pause in proliferation in the high JNK cells at 18 h and 24 h AHS, which was relieved by 36 h AHS (Fig. 3H-J), while the surrounding lower JNK cells did not show this pause (Fig. 3H-J). Thus, despite the different timing of ablation and recovery imposed by each ablation method, a similar pattern of wound signaling and proliferation is observed, while the acute injury caused by the DC system further reveals that cells have distinct responses based on JNK levels.

We next analyzed *DR$^{WNT}$-GFP* enhancer activity in this dynamic context (Fig. 3K-M). Early *DR$^{WNT}$-GFP* activity (0-12 h AHS) significantly overlapped with JNK (Fig. 1E), then expanded by 18 h AHS to include peripheral cells with lower JNK and new JAK/STAT activity (Figs 1F-F'''' and 3B,L). At 24 h AHS, DR$^{WNT}$ remained active in both central high JNK and peripheral JAK/STAT-positive regions, even as central JAK/STAT declined (Fig. 3M), likely due to repression by JNK (Jaiswal et al., 2023). Together, these results show that the DR$^{WNT}$ enhancer activation begins in JNK-positive cells and later spreads to include JAK/STAT-positive regions in the wound periphery. This dynamic overlap suggests that both pathways may contribute to DR$^{WNT}$ activity in a developing wound.

## JAK/STAT directly regulates DR enhancer activity in the wound periphery

To test whether JAK/STAT is required for DR enhancer activation, we knocked down *Stat92E* (*UAS-Stat92E$^{RNAi}$*) during regeneration and assessed *DR$^{WNT}$-GFP*. *Stat92E* knockdown significantly

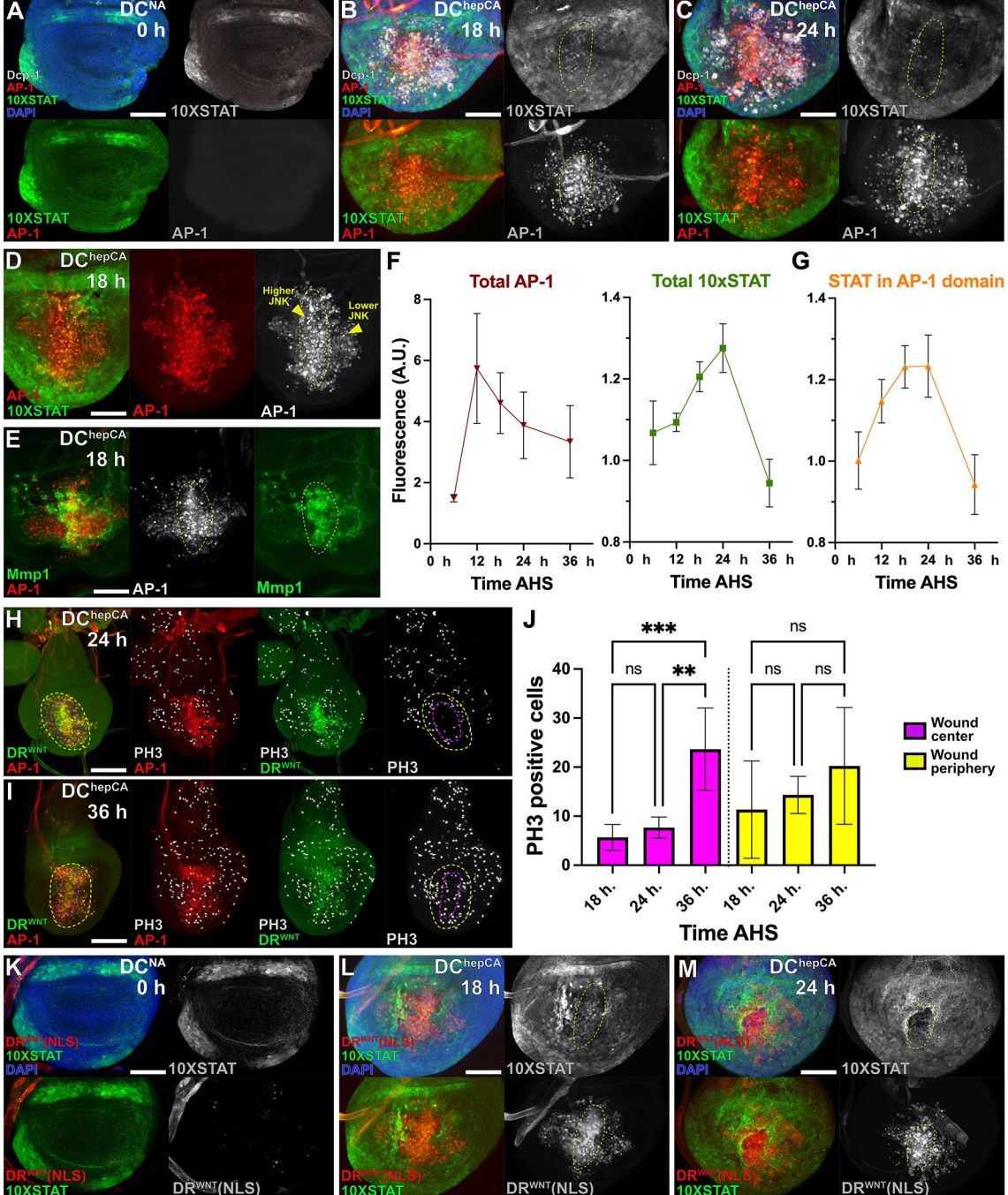

**Fig. 3. DR^WNT overlaps with dynamic changes in JNK and JAK/STAT expression during regeneration.** (A-C) Time course of discs ablated with DC^hepCA, imaged at 0, 18 and 24 h after heat shock (AHS), showing JNK activity (*AP-1-RFP*, red), JAK/STAT activity (*10×STAT-DGFP*, green) and cell death (Dcp-1, white). At 0 h, only developmental reporter expression is seen, with JAK/STAT restricted to the hinge. At 18 h, JNK is strongly induced in the wound center (dotted circle), and JAK/STAT is upregulated both within the wound and non-autonomously in surrounding tissue. By 24 h, JAK/STAT is reduced in the JNK-high wound center, while JNK remains elevated. (D) At 18 h AHS, discrete high and low JNK zones (red, dotted outlines) are visible in the distal pouch, coincident with JAK/STAT activity (yellow arrowheads). (E) Expression of JNK target Mmp1 (green) at 18 h highlights spatially distinct regions of high versus low JNK at 18 h AHS (dotted outline). (F) Quantification of JNK (AP-1-RFP) and JAK/STAT (10×STAT-DGFP) reporter intensity in the pouch, normalized to background, from 6-36 h AHS (*n*=21 discs; A.U., arbitrary units). (G) Quantification of JAK/STAT activity specifically within the JNK-positive domain (as in F), showing early co-activation followed by reduction of JAK/STAT signal in JNK-high regions and continued increase in peripheral regions (*n*=21). (H,I) Discs imaged at 24 h (H) and 36 h (I) AHS stained for proliferation (PH3, white), with JNK (red) and DR^WNT (green). At 24 h, high JNK regions (purple dashed outline) lack PH3-positive cells, while JNK-low periphery (yellow outline) shows ongoing proliferation. At 36 h, proliferation is restored in the JNK-high zone. (J) Quantification of PH3-positive cells within high and low JNK regions from 18-36 h AHS. High JNK regions show significantly reduced proliferation at 18 h and 24 h, recovering by 36 h (ordinary one-way ANOVA: 18 h versus 36 h ***P=0.0004, 24 h versus 36 h **P=0.006; *n*=23 discs). Proliferation in low JNK areas increases over time but differences are not statistically significant. (K-M) Discs expressing DR^WNT[NLS] (red) and *10×STAT-DGFP* (green), either unablated (DC^NA; K) or ablated (DC^hepCA; L,M). At 0 h AHS (K), no DR^WNT activity is observed, and JAK/STAT is limited to the hinge. At 18 h (L), DR^WNT overlaps with JAK/STAT in both the wound center and periphery. At 24 h (M), JAK/STAT decreases in the center while DR^WNT remains active (yellow dashed outlines). Data are mean±s.d. ns, not significant. Scale bars: 50 μm (D,E,H,I); 15 μm (A-C,K-M).

reduced $DR^{WNT}$-*GFP* activity throughout regeneration (Fig. 4A-B), as well as activity of other DR enhancers including $DRMS^{WNT}$-*GFP* and $DR^{Mmp1}$-*GFP* (Fig. S4A-G). Importantly, this reduction was most apparent in the wound periphery (Fig. 4A′), where normally JNK levels are low and JAK/STAT is high (Figs 1F and 3B-D,L), indicating that JAK/STAT is necessary for enhancer activation in this domain. *Stat92E* knockdown did not affect cell death (Fig. 4C,C′) or JNK signaling (Fig. 4D,D′), placing JAK/STAT function downstream or parallel to JNK. To assess the spatial requirement for JAK/STAT in the center of the wound versus the periphery, we knocked down *Stat92E* specifically in the posterior compartment using *hedgehog*-GAL4 (*hh-GAL4*), thus affecting only half the wound (Fig. 4E). $DR^{WNT}$ activity was mostly preserved centrally but was reduced in the posterior wound periphery (Fig. 4E), reinforcing the possibility that JAK/STAT is required for $DR^{WNT}$ activity at the wound periphery, where JNK is weaker.

The behavior of another DRMS enhancer identified in our accessibility dataset (Harris et al., 2020) was consistent with this idea. A GFP reporter for $DRMS^{TDC1/2}$, an enhancer located near to the *Tdc1* and *Tdc2* genes (Fig. S4H), which were both highly upregulated upon damage (Table S1), contained AP-1 but not Stat92E consensus sites (Fig. S4H), and exhibited DRMS behavior (Harris et al., 2020). $DRMS^{TDC1/2}$-*GFP* showed no activity in the periphery of a wound and was unaffected by *Stat92E* knockdown (Fig. S4I-J), suggesting it is JNK-dependent but JAK/STAT-independent. In this experiment, the minimal DR module for this enhancer has not been identified, therefore the full DRMS is used.

To test whether JAK/STAT acts directly on $DR^{WNT}$, we generated a mutant $DR^{WNT}$ reporter lacking its three identified consensus Stat92E binding sites ($DR^{WNT\Delta STAT}$-*GFP*; Fig. S4K). This reporter showed reduced activity in the wound periphery (Fig. 4F; Fig. S4L), phenocopying the *Stat92E* knockdown (Fig. 4A-D′), suggesting that JAK/STAT regulates the enhancer directly.

As noted, JAK/STAT is well established as necessary for regeneration (Katsuyama et al., 2015; La Fortezza et al., 2016; Herrera and Bach, 2019; Verghese and Su, 2016; Ahmed-de-Prado et al., 2018; Jaiswal et al., 2023; Floc'hlay et al., 2023), and indeed the knockdown of *Stat92E* following ablation by DC$^{hepCA}$ strongly limited regeneration in this system, as assayed by both wing scoring (Fig. 4G) and wing area measurements (Fig. 4H). Although JAK/STAT supports regeneration through pleiotropic effects, including promoting cell survival and proliferation (Herrera and Bach, 2019), these findings highlight an additional mechanism through which JAK/STAT could promote regeneration: by directly regulating the activity of DR enhancers and thus their targets.

### Ectopic JAK/STAT is insufficient to activate DR enhancers
Although JAK/STAT signaling is necessary for the full spatial activity of DR enhancers downstream of JNK, the question remains as to whether it is sufficient to induce their activity in the absence of JNK. To test this, we overexpressed *hop* in the distal pouch (*salm-GAL4, UAS-hop48a*) to ectopically activate JAK/STAT signaling. Although this expression activates $10\times STAT$-*DGFP* in the pouch (Fig. 5A), it is insufficient to activate the $DR^{WNT}$-*GFP* reporter (Fig. 5B). This is even despite the presence of low levels of cell death caused by *hop* overexpression (Fig. 5A,B). We repeated this experiment with different GAL4 drivers that express throughout the whole pouch (*rotund-GAL4*) or in both the pouch and the notum (*hh-GAL4* and *ptc-GAL4*). These drivers activated JAK/STAT signaling (Fig. S5A,C) and even led to the formation of ectopic pouch tissue (Fig. S5C,D) as described previously (Worley et al., 2018). However, they did not activate the $DR^{WNT}$ enhancer

(Fig. S5B,D) or other JNK targets such as *Mmp1* (Fig. S5A,B). The use of a stronger *hop* construct (*UAS-hopORF*) led to significant cell death, which in turn triggered JNK signaling shown by *Mmp1*, and thus JAK/STAT and $DR^{WNT}$-*GFP* expression (Fig. S5E,F). These data show that JAK/STAT signaling alone is not sufficient to activate DR enhancers, but rather is required to fully promote and maintain their activity after activation by JNK.

As JNK is a prerequisite for DR enhancer activation, we wondered whether JAK/STAT could alter DR enhancer behavior in different contexts where JNK is already present: either following damage or during normal development. We first tested ectopically expressing *hop* during ablation and found that additional JAK/STAT signaling did not significantly alter $DR^{WNT}$-*GFP* expression (Fig. 5C-E), suggesting that JAK/STAT is not limiting for DR enhancer activity during regeneration. Next, to examine whether JNK that occurs during development can activate $DR^{WNT}$-*GFP* when JAK/STAT is introduced, we expressed *hop* in the notum, where endogenous JNK promotes thorax closure and JAK/STAT is normally absent (Zeitlinger and Bohmann, 1999; Agnes et al., 1999; Martin-Blanco et al., 2000) (Fig. S1I). We used two different tissue-specific GAL4 drivers, *patched* and *pannier* (*ptc-GAL4* and *pnr-GAL4*), both of which expressed in domains that overlap JNK in the notum (Fig. 5F-G″; Fig. S5G,H) and activated the $10\times STAT$ reporter in response to *hop* (Fig. 5F″,G″). Even with the activity of both signaling pathways, the $DR^{WNT}$-*GFP* reporter failed to activate (Fig. 5H-I′), suggesting that JNK expressed during development is not sufficient to activate DR enhancers, even with the addition of JAK/STAT. By contrast, JNK during injury initiates DR enhancer activity, while the subsequent induction of JAK/STAT allows their expansion into the wound periphery. Since cell death and its associated downstream signaling are dispensable for DR enhancer activity (Fig. 1), this reduces the likelihood that injury-specific factors are responsible for this context-dependent activation and raises the possibility that JNK signaling intensity might instead regulate DR enhancer activation.

### A threshold level of JNK activity is required to initiate the regeneration program via DR enhancers
To test whether the level of JNK signaling influences DR enhancer activation, we developed a method to induce different intensities of steady-state JNK activity, unlike the dynamic changes that occur within a wound. Using the temperature sensitivity of GAL4 (Duffy, 2002), we expressed a wild-type *hep* transgene in the distal pouch (*salm-GAL4, UAS-hep*$^{WT}$) at 22°C for JNK$^{Low}$ and 30°C for JNK$^{Med}$, and a constitutively active *hep* (*salm-GAL4, UAS-hep*$^{CA}$, *GAL80*$^{ts}$) activated by temporary culture at 30°C for JNK$^{High}$. Cell death was blocked with *mir(RHG)* to reduce ROS and limit feedforward JNK (Santabarbara-Ruiz et al., 2015; Khan et al., 2017; Santabarbara-Ruiz et al., 2019). Indeed, ROS were only minimally produced under these conditions versus ablation (Fig. S6C-C″), and JNK activity was consequently confined to the distal pouch (Fig. 6C). Expression of the JNK target *mol* that is important for ROS production was also unchanged (Fig. S6E-F″). We confirmed different JNK activity levels using *AP-1-RFP* quantified via fluorescence intensity (Fig. 6A) and qRT-PCR of transcript expression (Fig. 6B).

With this setup, we assessed DR enhancers and known regeneration targets *Mmp1*, *Ilp8*, *wg* and *Stat92E* (Fig. 6C). JNK$^{Low}$ failed to activate $DR^{WNT}$-*GFP* or JAK/STAT signaling, although it did activate *puc*, which likely has a lower activation threshold. JNK$^{Med}$ induced weak $DR^{WNT}$ and $DR^{Mmp1}$ reporter activity, but not robust expression of other targets. By contrast, JNK$^{High}$ strongly activated DR enhancers, including $DR^{WNT\Delta STAT}$, and all tested regeneration genes within the high-JNK domain (Fig. 6C). Interestingly, JAK/STAT

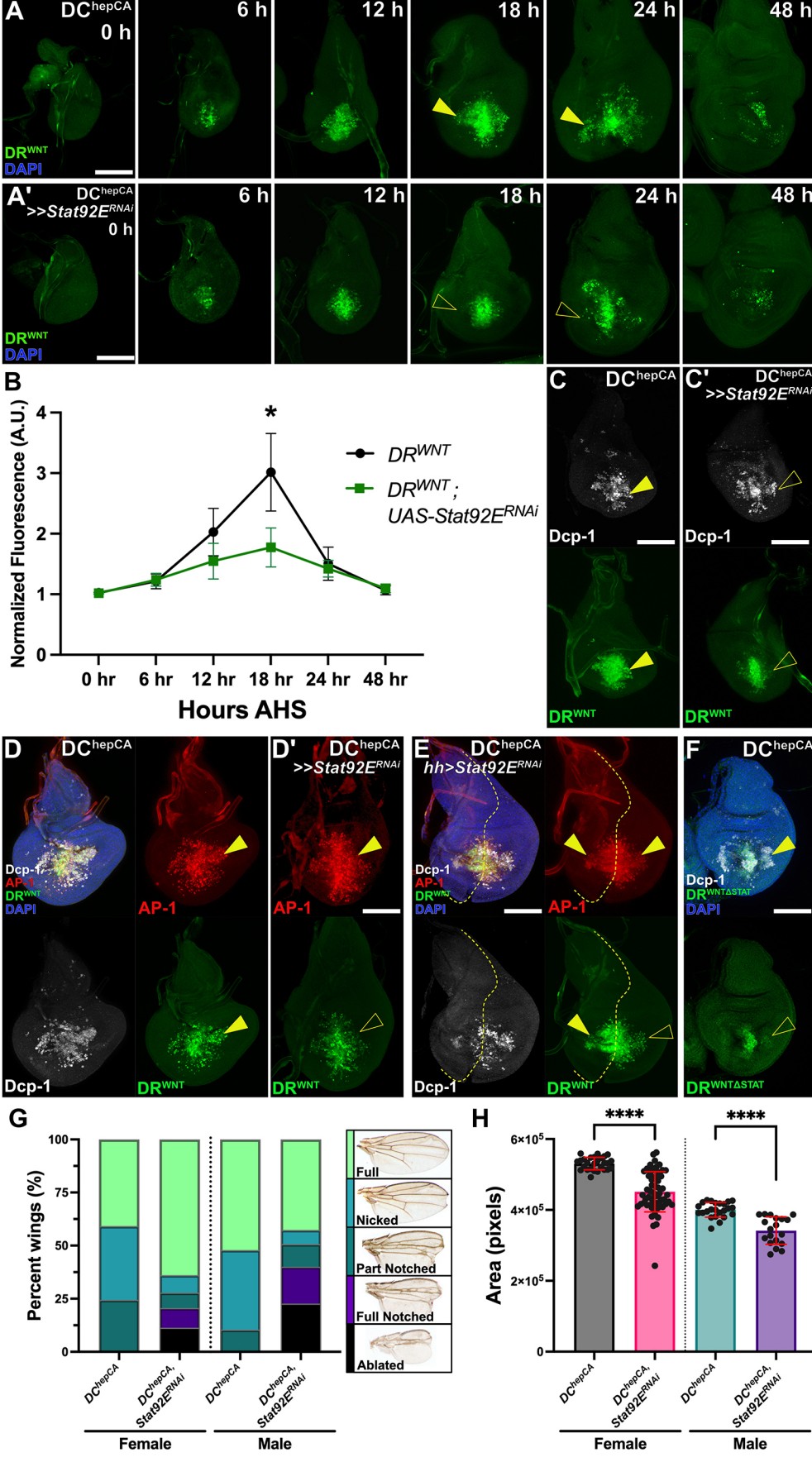

**Fig. 4. Stat92E is necessary for full activation and lateral domain activity of DR^WNT.** (A,A′) Time course (0-48 h AHS) of *DR^WNT-GFP* (green) reporter activation in DC^hepCA-ablated discs with (A) or without (A′) *Stat92E* knockdown. Peripheral DR^WNT activity is reduced at 18 and 24 h AHS with *Stat92E^RNAi* (arrowheads). (B) Quantification of DR^WNT fluorescence (normalized to background) confirms significant reduction at 18 h AHS with *Stat92E^RNAi*. Unpaired two-tailed *t*-tests: **P<0.000001; *n*=44 (control), *n*=45 (*Stat92E^RNAi*). A.U., arbitrary units. (C,C′) *DR^WNT-GFP* (green) and Dcp-1 (white) at 18 h AHS show similar cell death with or without *Stat92E*, but loss of peripheral DR^WNT (yellow arrowheads) in knockdown. (D-D′) Discs as in C,C′ with the JNK reporter (AP-1, red). The peripheral DR^WNT activity (yellow arrowhead) is lost upon *Stat92E* knockdown, while JNK activity is unaffected. (E) Posterior-specific *Stat92E* knockdown (*hh-GAL4*) shows reduced DR^WNT in posterior wound periphery (yellow arrowheads) compared to wild-type anterior; anterior/posterior boundary indicated by dashed line. (F) *DR^WNTΔSTAT-GFP* (lacking *Stat92E* sites) shows loss of peripheral activity at 18 h AHS (yellow arrowhead). (G) Adult wing phenotype scoring post-ablation shows impaired regeneration with *Stat92E^RNAi*; *n*=127 (control), *n*=172 (*Stat92E^RNAi*), separated by sex. (H) Wing size is reduced in adults with *Stat92E^RNAi*. Unpaired two-tailed *t*-tests: ****P<0.0001; *n*=51 (control), *n*=78 (*Stat92E^RNAi*), data shown by sex. Data are mean±s.d. Scale bars: 50 µm.

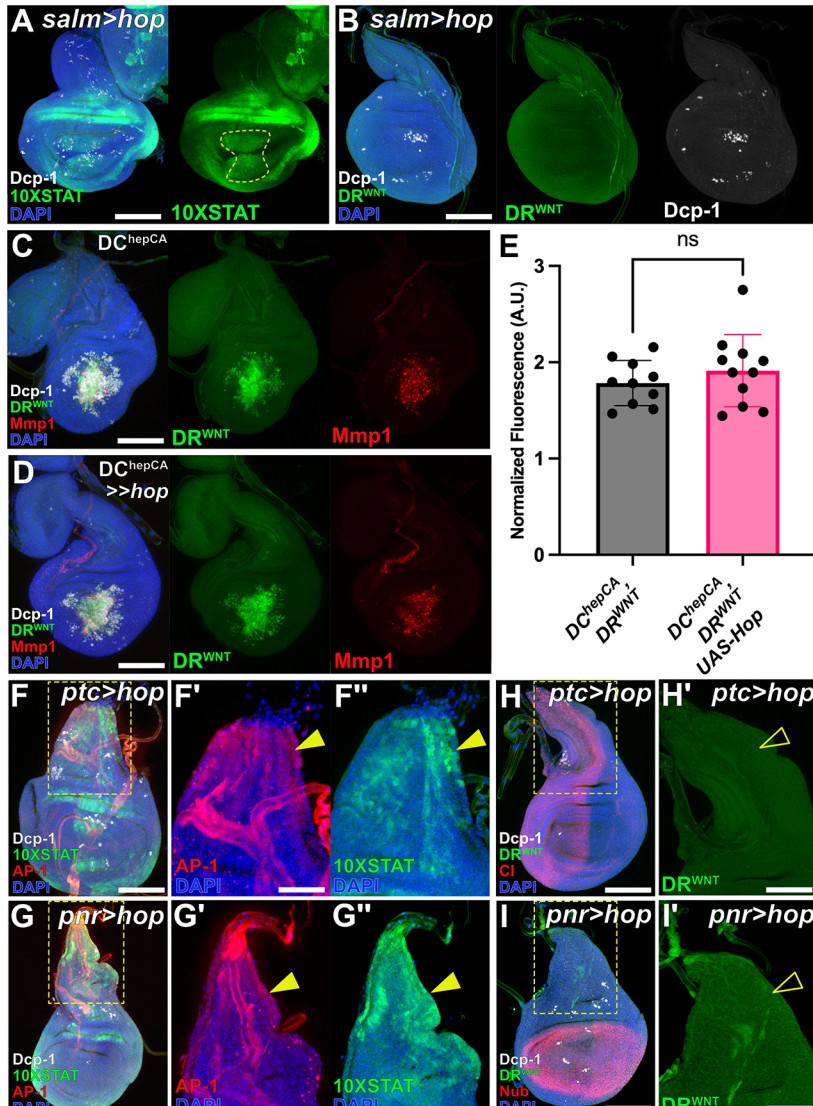

**Fig. 5. JAK/STAT activation is not sufficient to induce DR$^{WNT}$.** (A,B) Ectopic JAK/STAT signaling in the distal pouch (*salm-GAL4, UAS-hop*; yellow outline) activates *10×STAT-DGFP* (green in A) without inducing cell death (Dcp-1, white) or DR$^{WNT}$ reporter expression (green in B). (C,D) In DC$^{hepCA}$-ablated discs, DR$^{WNT}$ is activated (green) alongside Dcp-1 (white) and Mmp1 (red, JNK activity). Ectopic JAK/STAT (*UAS-hop*) does not alter DR$^{WNT}$ or JNK activity. (E) Quantification of DR$^{WNT}$ reporter intensity shows no significant difference with ectopic JAK/STAT. *n*=10 (DC$^{hepCA}$), *n*=11 (DC$^{hepCA}$, UAS-hop). Unpaired two-tailed *t*-test. A.U., arbitrary units. ns, not significant. Data are mean±s.d. (F-F″) Overexpression of *hop* in an anterior/posterior (A/P) stripe (*ptc-GAL4, UAS-hop*) increases *10×STAT-DGFP* (green) and JNK (AP-1-RFP, red) in the notum (arrowheads), without cell death (Dcp-1, white). (G-G″) Ectopic JAK/STAT in the notum (*pnr-GAL4, UAS-hop*) similarly elevates STAT and JNK reporters (green and red), without cell death. (H,H′) DR$^{WNT}$ reporter (green) is not activated by A/P stripe JAK/STAT (open arrowhead in H′); Ci (red) marks the A/P boundary, Dcp-1 (white) shows no cell death. (I,I′) Ectopic JAK/STAT in the notum (*pnr-GAL4, UAS-hop*) also fails to activate DR$^{WNT}$ (green; open arrowhead in I′). Nub (red) marks the pouch, Dcp-1 (white) labels apoptosis. Right panels in F-I′ show magnified views of boxed areas in left panels. Scale bars: 50 μm (A-D,F,G,H,I); 25 μm (F′,F″,G′,G″,H′,I′).

activation occurred outside this domain (Fig. 6D), consistent with the non-autonomous induction observed in wounds (Jaiswal et al., 2023), but did not lead to DR enhancer activation there, likely because *mir(RHG)* prevents JNK from spreading and forming a low-JNK domain needed for co-activation with JAK/STAT. Thus, these experiments demonstrate three important findings: (1) a threshold level of JNK is needed to activate JAK/STAT signaling, which in steady-state conditions (and a mature wound) becomes exclusively non-autonomous outside of the area of JNK activity; (2) at high levels of JNK, DR enhancers and their downstream pro-regeneration targets can be activated in the absence of JAK/STAT; and (3) when the spread of JNK and the subsequent formation of different levels is prevented, DR enhancer activation outside of the high JNK domain does not occur, even in the presence of JAK/STAT.

Finally, to confirm the requirement for lower JNK combined with JAK/STAT to promote normal DR enhancer behavior in a wound context, we observed manipulations of these factors in DC$^{hepCA}$ ablation (Fig. 7A-A‴). Knockdown of *Stat92E* inhibits the spread of DR enhancer activity into the wound periphery where JNK is lower (Fig. 7B-B‴) but does not affect ROS production (Fig. S6D′) or JNK levels (Fig. 7B-B‴). Blocking cell death with *mir(RHG)* to prevent the formation of different JNK levels within the wound

restricted *DR$^{WNT}$-GFP* to the high JNK domain (Fig. 7C-C‴). Combining *mir(RHG)* and *Stat92E$^{RNAi}$* phenocopies this result (Fig. 7D-D‴). Together, these results show that DR enhancer activation during regeneration requires either high JNK at the wound center or the combination of low JNK and JAK/STAT in the periphery.

Overall, these data support a model in which spatially distinct thresholds of JNK regulate regenerative gene expression (Fig. 7E-H′). High JNK in the center of the wound is sufficient to activate DR enhancers (Fig. 7E,E′). This high JNK also induces JAK/STAT signaling, which later becomes repressed in these same cells by JNK but activated non-autonomously in adjacent low JNK tissue (Fig. 7F,F′). In this peripheral domain the combination of JAK/STAT and low JNK co-activate DR enhancers (Fig. 7F,F′). When JAK/STAT is removed, DR enhancers activity in this region is lost (Fig. 7G,G′). Similarly, when cell death is blocked by *mir(RHG)*, limiting the spread of JNK beyond the center of the wound, DR enhancer activity becomes confined to the high JNK domain (Fig. 7H,H′). This mechanism ensures that DR enhancers respond proportionally to injury: high JNK initiates the response, while JAK/STAT extends enhancer activation outward. As DR enhancers likely regulate diverse genes (Harris et al., 2020), these JNK thresholds

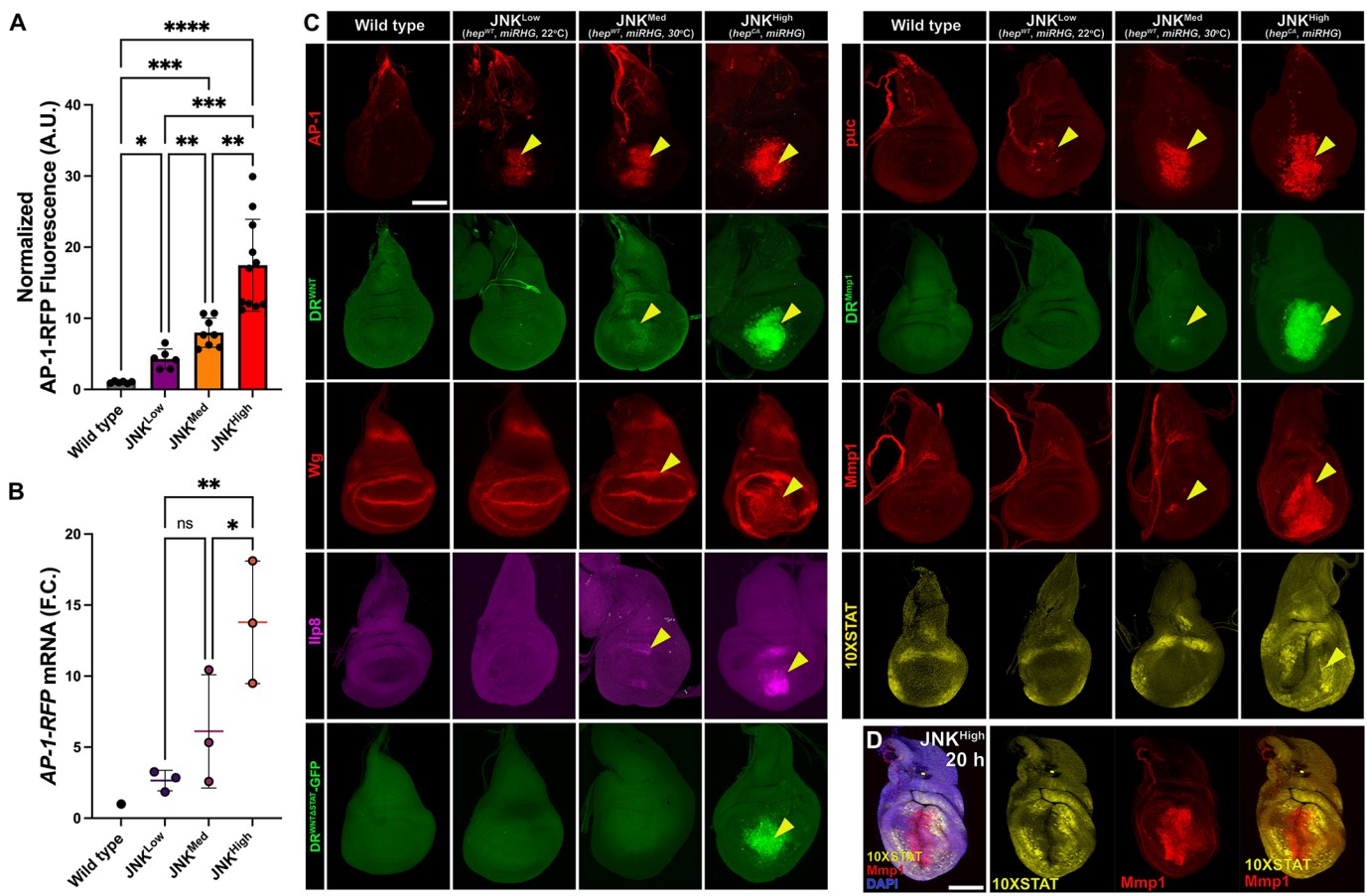

**Fig. 6. A threshold level of JNK signaling is required to activate DR^WNT^ and downstream targets.** (A) Quantification of JNK levels (AP-1-RFP fluorescence, normalized to background) in JNK^Low^ (salm>hep^WT^, 22°C), JNK^Med^ (salm>hep^WT^, 30°C) and JNK^High^ (salm>hep^CA^) discs, all with blocked apoptosis (*UAS-mir[RHG]*). One-way ANOVA: *$P$=0.0108 (WT–JNK^Low^), **$P$<0.0031 (JNK^Med^–JNK^High^), **$P$=0.0097 (JNK^Low^–JNK^Med^), ***$P$=0.0002 (WT–JNK^Med^ and JNK^Low^–JNK^Med^), ****$P$<0.0001 (WT–JNK^High^). $n$=6 (WT), 7 (JNK^Low^), 8 (JNK^Med^), 11 (JNK^High^); A.U.=Arbitrary Units. (B) Measurement of JNK levels in conditions of JNK^Low^, JNK^Med^ and JNK^High^ (as above) via qRT-PCR of *AP-1-RFP* transcript levels. Graphs show the mean fold change (F.C.) of *AP-1-RFP* in each condition relative to the wild-type (physiological level of *AP-1-RFP* in uninjured discs at the specified temperature). Ordinary one-way ANOVA between JNK conditions: *$P$<0.033 (JNK^Med^–JNK^High^), **$P$=0.0072 (JNK^Low^–JNK^High^), ns, not significant (JNK^Low^-JNK^Med^). $n$=3 independent biological repeats per condition. (C) Reporter activation under increasing JNK levels. AP-1-RFP and *puc-lacZ* (JNK reporters) activate at all levels (yellow arrowheads). DR^WNT^, DR^Mmp1^, Mmp1 and Ilp8 activate weakly at JNK^Med^ and strongly at JNK^High^. 10×STAT reporter is activated only at JNK^High^. (D) Under JNK^High^ conditions, *10×STAT-DGFP* (yellow) and the JNK target Mmp1 (red) become spatially distinct, emulating a mature-stage wound. Data are mean±s.d. Scale bars: 50 μm.

provide a spatial framework for controlling the magnitude, timing and domain of regenerative gene expression, and distinguish between the activation of targets required for regeneration versus normal development.

## DISCUSSION

Our previous work identified and characterized DRMS enhancers; regulatory elements within the *Drosophila* genome that activate regeneration-promoting genes in response to damage, but become silenced with maturity to limit tissue repair (Harris et al., 2016, 2020). These enhancers are activated by JNK signaling via a DR region, although it is unclear why this activation is injury-specific and is absent from other JNK-dependent contexts like embryogenesis or pupariation (Zeitlinger and Bohmann, 1999; Agnes et al., 1999; Harden, 2002; Kiehart et al., 2017). It is also unknown how a single pathway like JNK produces spatially dynamic gene expression during regeneration.

We have addressed both issues here. Firstly, by isolating JNK signaling from the downstream events of apoptotic signaling, we have shown that DR enhancers activate independently of cell death,

requiring only JNK and its immediate targets. One such target, JAK/STAT, is an additional direct input into enhancer activity alongside JNK. We find that high JNK levels are sufficient to initiate DR enhancer activity, while JAK/STAT amplifies and expands this activity into wound-proximal regions where JNK alone is insufficient. Below this threshold, only certain JNK targets like those necessary during development (*puc*), are expressed.

Within a wound, apoptosis promotes ROS-mediated P38 and high JNK activity in surrounding cells (Santabarbara-Ruiz et al., 2015; La Marca and Richardson, 2020), activating DR enhancers and their targets (e.g. *wg*, *Mmp1*), alongside promoting a transient senescence-like state (Jaiswal et al., 2023). JNK also induces expression of upd ligands, activating JAK/STAT signaling (Santabarbara-Ruiz et al., 2015; Katsuyama et al., 2015; La Fortezza et al., 2016; Ahmed-de-Prado et al., 2018; Pastor-Pareja et al., 2008; Worley et al., 2018), which initially acts in an autocrine fashion within the wound, but subsequently becomes paracrine in the wound periphery with low JNK (Jaiswal et al., 2023). In these outer regions, JAK/STAT and JNK enable DR enhancer activity, thus extending and sustaining regeneration beyond the initial wound. When

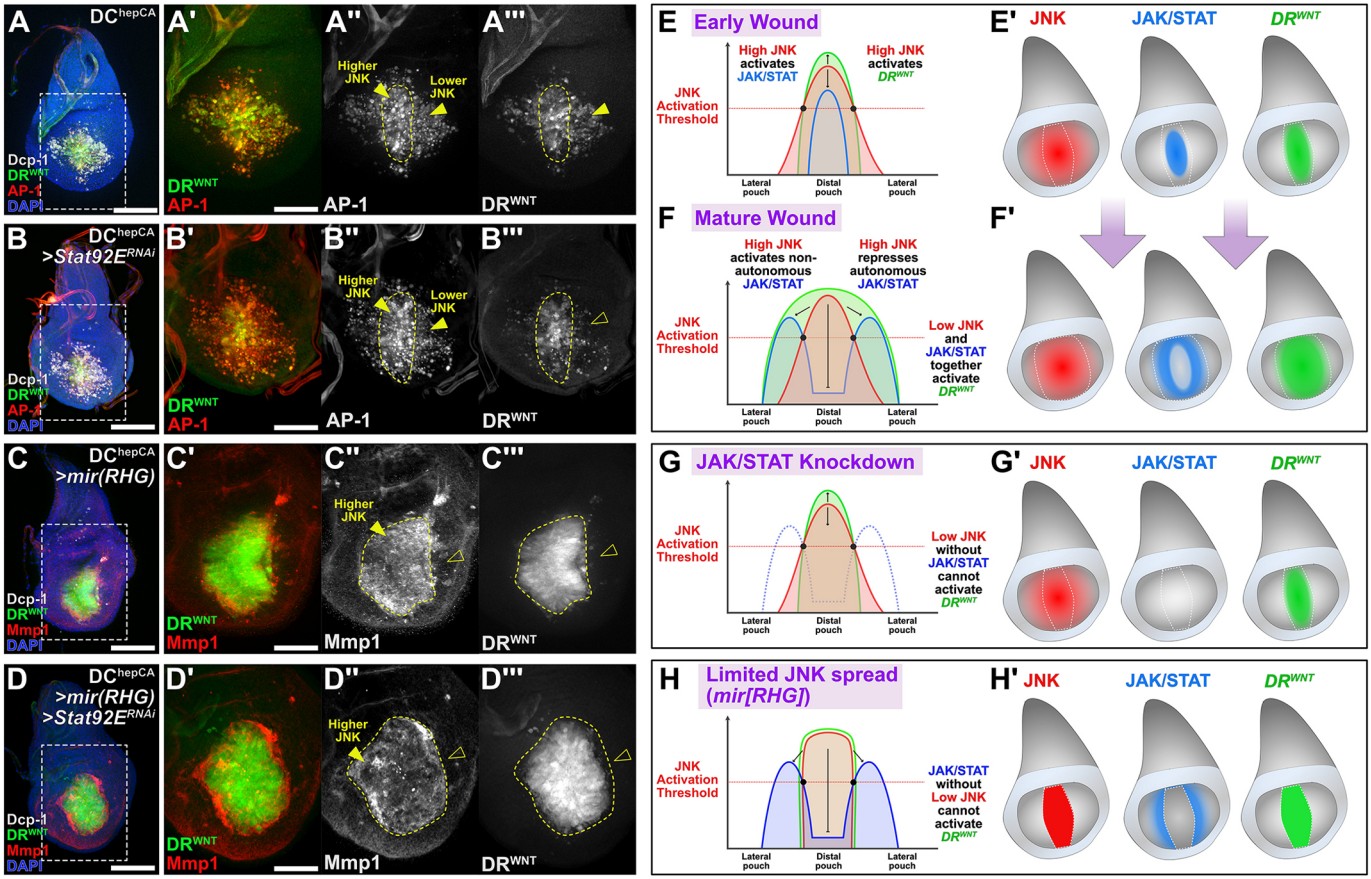

**Fig. 7. JNK and JAK/STAT expression dynamically activates DR$^{WNT}$ during wound formation.** (A-D‴) Discs bearing the DR$^{WNT}$ reporter (green) and the AP-1 reporter (red in A and B) or stained for JNK target Mmp1 (red in C and D), ablated by DC$^{HepCA}$. Discs are wild type (A), with *Stat92E* knockdown (B), with cell death blocked by *mir(RHG)* (C) or with *Stat92E* and cell death block by *mir(RHG)* together (D). DR$^{WNT}$ activity occurs in cells of both high and low JNK of a wound (A,A′), while lack of JAK/STAT limits DR$^{WNT}$ activity to cells with high JNK (B-B‴). Blocking cell death via *mir(RHG)* to inhibit ROS-mediated spread of JNK prevents the formation of a low JNK area where JAK/STAT can be activated, and therefore DR$^{WNT}$ is limited to the high JNK cells in the center of the distal pouch (C-C‴). Preventing the spread of JNK via blocking cell death with *mir(RHG)* and blocking JAK/STAT shows the same lack of DR$^{WNT}$ activity outside of the distal pouch (D-D‴). Limited JNK spread is shown by Mmp1 staining in C and D. Yellow dashed outlines indicate area of high JNK. Yellow arrowheads indicate high JNK cells in the center of the wound and lower JNK cells in the wound periphery. Open arrowheads indicate lack of DR$^{WNT}$ activity in B‴,C‴ and D‴, and lack of JNK activity shown by Mmp1 in C″ and D″. (E-H′) Graphical representation of the mechanism by which JNK and JAK/STAT expression leads to DR$^{WNT}$ activity across a developing wound in early L3 larvae from the initial injury (up to ~18 h AHS) to the mature injury (after ~18 h AHS) (E-F′), and in conditions of JAK/STAT knockdown (G,G′) or when JNK signaling spread is prevented (H,H′). See main text for details. Right panels in A-D‴ show magnified views of boxed areas in left panels. Scale bars: 50 μm (A,B,C,D); 25 μm (A′-A‴,B′-B‴,C′-C‴).

JAK/STAT is inhibited or JNK is spatially restricted by blocking cell death signaling, peripheral enhancer activation fails, limiting the regeneration program to the immediate wound site. Thus, DR enhancers not only initiate regenerative gene expression downstream of JNK but also translate heterogenous levels of JNK into dynamic spatial and temporal expression patterns required for regeneration.

## DRMS enhancers as a mechanism to spatially regulate regenerative gene expression

Regeneration-specific enhancers have been identified in diverse organisms and tissues (Rodriguez and Kang, 2020; Harris, 2022; Yang and Kang, 2019). In highly regenerative zebrafish, such elements at the *leptin B* and *ctgfa* genes are involved in heart and fin regeneration (Kang et al., 2016; Pfefferli and Jazwinska, 2017), with numerous other damage-responsive loci found genome-wide (Wang et al., 2020; Goldman et al., 2017). Such regulatory elements likely function in other injury models, potentially even in humans (Lander et al., 2017; Suzuki et al., 2019). Their existence aligns with the need for regeneration-specific expression patterns, such as those that

are unique to wound healing or that bridge wound healing to developmental programs. Indeed, genome-wide studies in axolotl, *Xenopus*, mouse and *Drosophila* regeneration confirm the existence of such transcriptional and epigenetic profiles (Floc'hlay et al., 2023; Poss, 2010; Gerber et al., 2018; Aztekin et al., 2019; Storer et al., 2020; Worley et al., 2022).

Our work in imaginal discs suggests that regeneration enhancers do more than initiate reparative gene expression – they also reinforce heterogeneity in the regenerative response. A single pathway, JNK, simultaneously activates multiple genes, enabling a rapid and coordinated response to damage. However, differences in JNK levels caused by proximity to injury or tissue identity (Martin and Morata, 2018), must be decoded to produce appropriate spatial gene expression. DR enhancers enable this by translating graded JNK activity into differential target gene expression needed for recovery. Comparisons of full versus partial disc ablation (Harris et al., 2016, 2020), or clonal patches of varying size that are triggered to die (Harris et al., 2016), support the idea that DR enhancers drive spatial expression diversity based on JNK intensity. Single cell sequencing

of ablated wing discs also corroborates this (Floc'hlay et al., 2023; Worley et al., 2022), showing that cell populations with discrete cellular identities form following injury and change over time. Injuries within the wing pouch become striated through the formation of a central JNK-expressing population, which downregulates markers for proliferation and expresses factors like *Ets21C* to limit differentiation, alongside upregulation of paracrine signaling factors like *wg* and the upd ligands (Jaiswal et al., 2023; Floc'hlay et al., 2023; Worley et al., 2022). Cells of the periphery receive these signals, gain JAK/STAT activation and express factors that support survival and proliferation (Jaiswal et al., 2023; Floc'hlay et al., 2023; Worley et al., 2022). This regionalization, originally stemming from JNK signaling, is crucial for coordinating growth and repatterning during regeneration (Jaiswal et al., 2023), facilitated by DR enhancers.

### Signaling factors that activate and regulate DR enhancers are potentially conserved

JNK and JAK/STAT signaling are both essential for regeneration in *Drosophila*, with complex roles in proliferation, cell survival and identity (Pinal et al., 2019; Herrera and Bach, 2019; Katsuyama et al., 2015; La Fortezza et al., 2016; Herrera and Bach, 2019; Verghese and Su, 2016; Ahmed-de-Prado et al., 2018; Jaiswal et al., 2023). In many cases, JAK/STAT in the context of a wound depends on prior JNK activity, raising the possibility that DR enhancers could mediate any events in regeneration that rely on input from both pathways. JNK is closely associated with regeneration enhancers in many if not all known examples, even in organisms with highly contrasting abilities to regenerate, like zebrafish and mammals (Kang et al., 2016; Wang et al., 2020; Thompson et al., 2020; Guenther et al., 2015; Aguilar et al., 2016), and in diverse tissues, from imaginal discs to Schwann cells (Harris et al., 2016, 2020; Vizcaya-Molina et al., 2018; Arthur-Farraj et al., 2017), implying it is likely a conserved input (Harris, 2022). JAK/STAT, though not yet linked to DR enhancers in other species, is essential for regeneration in many contexts. For example, zebrafish fin regeneration is impaired without *Stat3* due to aberrations in macrophage immune cell recruitment (Sobah et al., 2023). Thus, it will be important to test whether JAK/STAT plays a similar direct role in DR enhancer regulation outside *Drosophila*. Of course, additional signals may also refine enhancer activity as JAK/STAT does, although candidate screens in our lab have yet to identify such a signal. Irrespective of other pathways, identifying JAK/STAT as a novel DR enhancer input helps to further define the regeneration enhancer code, which may aid in their identification, and that of gene regulatory networks (GRNs) related to injury repair in *Drosophila* and beyond. A notable example of a mapped regeneration GRN is in the acoel worm, where Egr acts as a master epigenetic regulator to active numerous genes in response to amputation injury (Gehrke et al., 2019). Here again, JNK signaling is implicated in activating genes proximal to damage-responsive regions regulated by Egr (Gehrke et al., 2019). As DRs enhancers in imaginal discs are also epigenetically regulated (Harris et al., 2016, 2020), it is possible that a similar regulatory situation might exist in which pioneer factors are required to regulate the regeneration program via DR enhancers. The best characterized pioneer factors in *Drosophila* are encoded by *grainy head* (*grh*) and *zelda* (*zld*), the latter of which has recently been shown to be important for maintaining cell fate identity in regenerating wing discs (Bose et al., 2025). Exploring their roles may further clarify how DR enhancers regulate regeneration in *Drosophila*.

### Establishing a JNK threshold necessary for DR enhancer activation

JNK signaling plays a central role in regeneration, with different levels or duration of signaling producing distinct outcomes (Pinal et al., 2019; La Marca and Richardson, 2020); low or transient JNK promotes proliferation and survival, while higher or sustained JNK leads to cell death (Santabarbara-Ruiz et al., 2019). These outcomes may depend on whether signaling is autocrine or paracrine (Pinal et al., 2019), and on the use of distinct JNK receptors. While the receptor Grindlewald (Grnd) is activated by the ligand Egr to trigger high JNK and cell death, Wengen (Wgn) is activated by ROS to promote cell survival and proliferation via P38 and potentially lower JNK levels (Esteban-Collado et al., 2024). Thus, both signaling modality and receptor use may shape JNK's effects.

Our data show that specific JNK activity levels determine downstream target gene expression. One open question is how high JNK can selectively induce JAK/STAT, enabling expression of targets like *wg* and *Mmp1* in areas where JNK alone is insufficient. An explanation could lie in the involvement of other factors. For example, P38 is activated alongside JNK in regenerating cells, which together drive JAK/STAT activity via upd expression (Santabarbara-Ruiz et al., 2015). Alternatively, DR enhancers may regulate components of the JAK/STAT pathway themselves; indeed we identified damage-responsive regions adjacent to two upd genes (Harris et al., 2020), suggesting an autoregulatory loop of JAK/STAT signaling, akin to that of *dome* receptor feedforward expression observed in embryogenesis (Hombria et al., 2005).

Although only recently discovered, DR enhancers have already revealed how regeneration-associated gene expression is activated, patterned and resolved. As our work here shows, these enhancers can be injury-specific and translate a single damage input into the spatially dynamic patterns of gene expression required for proper regeneration, shutting off once regeneration is complete. This makes them powerful tools to manipulate repair processes, an idea that has already led to methods driving expression of endogenous and exogenous factors to enhance regeneration in various organisms (Harris et al., 2016; Kang et al., 2016), including adult mammals (Yan et al., 2023). Continuing the study of these enhancers is therefore essential to advance our understanding of regenerative biology.

### MATERIALS AND METHODS
#### *Drosophila* stocks
Flies were cultured in conventional dextrose fly media at 25°C with 12 h light/12 h dark cycles. Genotypes for each figure panel are listed in Table S3. Fly lines used as ablation stocks are as follows: hs-FLP; hs-p65; salm-LexADBD, DVE≫GAL4 (DC$^{NA}$), hs-p65/CyO; salm-LexADBD, DVE≫GAL4/TM6B, Tb (DC$^{hepCA}$) (Harris et al., 2020). Ablation stocks used for RNA-seq were previously outlined in Harris et al. (2020). DR/DRMS reporters were previously described in Harris et al. (2016, 2020). The following stocks were obtained from Bloomington *Drosophila* Stock Center (BDSC): AP-1-RFP (BL#59011), puc-lacZ (BL#11173), 10×STAT-DGFP (BL#26200, BL#26199), 10×STAT-GFP (BL#26197), UAS-Stat92E$^{RNAi}$ (BL#33637), hh-GAL4 (BL#600186), Pnr-GAL4 (BL#3039), rn-GAL4 (BL#7405), Ptc-GAL4 (BL#44612), UAS-Hep$^{CA}$ (BL#6406) UAS-Hep$^{WT}$ (BL#3908), salm[GMR85E08]-GAL4 (BL#46804), mol-lacZ (BL#12173). UAS-mir(RHG) was a gift from Iswar Hariharan at University of California, Berkeley, USA (Siegrist et al., 2010); UAS-hop48a (UAS-hop) and UAS-hop$^{ORF}$ were gifts from Erika Bach at New York University. Stocks generated for this work were: DR$^{WNT}$ΔStat92E (II), DR$^{WNT}$-RFP (II), DR$^{WNT}$-GFP (III), DR$^{WNT}$[NLS] (III). The DR$^{Wnt}$ΔSTAT92E transgenes were generated using In-Fusion PCR mutagenesis (Clontech) to sequentially delete the consensus sequences (primers listed in Table S3). The

DR^{WNT}-RFP and DR^{WNT}[NLS] transgenes were generated by replacing the *eGFP* coding sequence in the DR^{WNT}-eGFP (formerly BRV-B-GFP; Harris et al., 2016) reporter construct with the *DSred* coding sequence (Flybase ID: FBto0000019) from TRE-RFP (BDSC #59011) (Chatterjee and Bohmann, 2012) or Redstinger from pQUASp-RedStinger (nlsRFP) generated by Christopher Potter (Addgene plasmid #46165). Cloning primers for constructs are listed in Table S3. The *hsp70* promoter used in all reporter constructs comprises the minimal promoter sequence with the heat shock elements removed (Amin et al., 1988) to prevent induction by the heat shock used in ablation experiments. Reporters were inserted into the *AttP40* landing site via PhiC31 recombination, DR^{WNT}-eGFP (III) and DR^{WNT}[NLS] (III) were inserted in the VK00027 (BDSC #9744) cyto-location. Transgenic services were provided by BestGene.

### Ablation experiments
#### DUAL Control ablation
DC experiments were performed essentially as described in Harris et al. (2020). Briefly, experimental crosses were cultured at 25°C and density controlled at 50 larvae per vial. Larvae were heat shocked on day 3.5 of development for early L3 [84 h after egg deposition (AED)] or day 4.5 for late L3 (108 h AED) by placing vials in a 37°C water bath for 45 min, followed by a return to 25°C. Larvae were allowed to recover for 18 h before being dissected, fixed and immunolabeled, unless otherwise indicated. All discs examined by immunofluorescence were early L3 discs ablated and imaged at the indicated hour AHS, with the exception of late L3 discs shown in Fig. 1B. *UAS-y^{RNAi}* were used as control lines for RNAi-based experiments.

#### GAL4/UAS ablation
GAL4/UAS-based ablation experiments were performed essentially as described in Harris et al. (2020). Briefly, larvae bearing rn-*GAL4, GAL80^{ts}, UAS-egr* were cultured at 18°C and density controlled at 50 larvae per vial. Larvae were upshifted on day 7 (168 h AED) or day 9 (216 h AED) of development for 40 h at 30°C and dissected into Trizol immediately and prepared for RNA-seq.

#### Inducing JNK at different levels
Flies of genotype *UAS-hep^{WT}, UAS-mir(RHG), salm[GMR85E08]-GAL4* were used for JNK^{Low} and JNK^{Med} conditions. Crosses were density controlled at 50 larvae per vial and kept at either 22°C (JNK^{Low}) or 30°C (JNK^{Med}) until mid L3 (120 h and 88 h AED, respectively) before being dissected, fixed and stained. Flies of genotype *UAS-hep^{CA}, UAS-mir(RHG), GAL80^{ts}, salm[GMR85E08]-GAL4* were used for the JNK^{High} condition. Crosses were density controlled at 50 larvae per vial and kept at 18°C until day 6 (144 h AED) when they were transferred to 30°C for 20 h before being dissected, fixed and stained.

### Immunohistochemistry
Larvae were dissected in 1× PBS followed by a 20 min fix in 4% paraformaldehyde in PBS (PFA). After three washes in 0.1% PBST (1× PBS+0.1% Triton X-100), larvae were washed in 0.3% PBST and then blocked in 0.1% PBST with 5% normal goat serum (NGS) for 30 min. Primary staining was carried out overnight at 4°C, and secondary staining was carried out for 4 h at room temperature. The following primary antibodies were obtained from the Developmental Studies Hybridoma Bank: mouse anti-Nubbin (1:25, 2D4), mouse anti-Wg (1:100, 4D4), mouse anti-Mmp1 C-terminus (1:100, 14A3D2), mouse anti-Mmp1 catalytic domain (1:100, 3A6B4), mouse anti-LacZ (1:100, 40-1a), rat anti-Ci (1:3, 2A1) and rat anti-DE-cadherin (1:100, DECAD2). Antibodies obtained from Cell Signaling Technologies were: rabbit anti-Dcp-1 (1:1000, 9578), mouse anti-PH3 (1:500, 9706). Secondary antibodies were obtained from Invitrogen and used at a 1:500 dilution: anti-rabbit 647 (A-21244) and anti-mouse 555 (A-21422) and anti-rat 555 (A-21434). DAPI (1:1000, 4083) was used as a counterstain. Images were obtained on a Zeiss AxioImager M2 with ApoTome. For each experiment at least 15 discs were analyzed before choosing a representative image, and the number used for quantification is indicated in the figure legends (N). Images were processed using Affinity Photo and Affinity Designer. All images are set to the scale bar being 50 μm in length unless otherwise stated in the figure legend.

### DHE staining
Dihydroethidium (DHE) labeling was performed by incubating freshly dissected wing imaginal discs in Schneider's Medium with 1 μl of 10 mM DHE reconstituted in 1 ml DMSO (for a working concentration of 10 μm DHE) for 10 min, followed by three 5 min washes in PBS, fixed in 4% PFA for 8 min, a further 1 min wash in PBS, followed by mounting and immediate imaging.

### Regeneration scoring and wing measurements
Wings of adult flies from heat-shocked larvae were scored and measured after genotype obscuring by another researcher. Scoring was performed on anesthetized adults by binning into a regeneration scoring category (Harris et al., 2020; Klemm et al., 2021). Wing measurements were performed by removing wings, mounting in Permount solution (Thermo Fisher Scientific) and wings imaged using a Zeiss Discovery V8 microscope. Wing area was measured using the Fiji software. Male and female adults were measured separately to account for sex differences in wing size using a reproducible measuring protocol that excludes the variable hinge region of the wing (details of measuring protocol available on request). Statistics were performed using GraphPad Prism 10.0.

### Quantification, fluorescence measurement and statistical analysis
Adult wings, mean fluorescence intensity and cell counts were measured using Fiji. GraphPad Prism 10.0 was used for statistical analysis and graphical representation. Graphs depict the mean of each treatment, while error bars represent the standard deviation. The mean fluorescence intensity (MFI) was quantified in Fiji using arbitrary units (A.U). In each experiment, MFI of the wing pouch was normalized to the MFI of the non-fluorescent notum. For measurements of AP-1 intensity (Fig. 1G), the profile tool on the Zen Imaging platform was used to generate data points across injured discs, which was standardized across disc widths by converting to percent distance, and analyzed in Prism. For measurement of STAT in the AP-1 domain (Fig. 3G), the area of the AP-1 expression domain was selected and the MFI of 10×STAT-DGFP in this domain was measured and normalized. For measurement of PH3 cell number (Fig. 3H,I), cells were counted using the ImageJ cell counter plug-in. The wound center and periphery areas were selected by outlining the areas of high and low JNK expression using the AP-1 reporter, and the cells in each area counted. For all quantifications the mean and standard deviation for each normalized treatment was calculated and used for statistical analysis. The sample size and *P* values for all statistical analyses are indicated in the figure legends. Statistical significance was evaluated in Prism 10.0 using the statistical test stated.

### qRT-PCR analysis
Total RNA was extracted from 15 wing discs per sample using the TRIzol reagent (Thermo Fisher Scientific), and cDNA libraries were prepared from total RNA using the AzuraQuant cDNA synthesis kit (Azuragenomics) according to the manufacturer's instructions. qRT-PCR was performed using AzuraQuant Green 1-step qPCR Mix LoRox (Azuragenomics) on a Thermo Fisher Quant Studio3. Data were analyzed using the ΔΔCt method and normalized to the housekeeping gene *Act5C*. Primer sequences used for PCR: *Act5C* Fwd GGCGCAGAGCAAGCGTGGTA, Rvse GGGTGCCA-CACGCAGCTCAT; *DsRed* Fwd CACTACCTGGTGGAGTTCAAG, Rvse GATGGTGTAGTCCTCGTTGTG.

### RNA-seq
Library prep and RNA-seq was performed by the QB3 Genomics Facility at University of California, Berkeley, from total RNA of ~100 discs per sample using polyT adapters and sequenced on the Illumina Miseq platform.

### Data processing
The RNA-seq data was processed using a pipeline that included quality control, trimming, read alignment and gene expression quantification. Adapters and low-quality reads were trimmed using Trim Galore (http://www.bioinformatics.babraham.ac.uk/projects/trim_galore) using the Phred quality score of 20. We visualized quality of the raw and trimmed reads using FastQC and MultiQC (Ewels et al., 2016). Trimmed reads were

aligned to the *Drosophila melanogaster* reference genome obtained from FlyBase (Flybase.org; version r6.57) using HISAT2 (Kim et al., 2019). The resulting BAM files were sorted, and coverage statistics were calculated using SAMtools (Li et al., 2009). Finally, gene-level read counts were obtained using featureCounts based on the *Drosophila melanogaster* annotation (version r6.57), also obtained from FlyBase (Liao et al., 2014).

### Differential expression

Differential gene expression analysis was performed using the limma package in R (Smyth, 2005) with the voom transformation (Law et al., 2014) applied to account for the mean-variance relationship in RNA-seq data. After transforming the read counts into $LOG_2$-counts per million ($LOG_2$-CPM), a linear model was fitted to the data to estimate the differential expression between early and late L3 stages in damaged and undamaged conditions. Empirical Bayes moderation was applied to the standard errors to improve accuracy, and *P*-values were adjusted for multiple testing using the Benjamini-Hochberg method. Genes with an adjusted *P*-value less than or equal to 0.05 and an absolute $LOG_2$ fold change ($LOG_2FC$) greater than or equal to 2 were considered differentially expressed.

### Gene enrichment analysis

A gene ontology (GO) enrichment analysis was performed using the clusterProfiler package in R (Wu et al., 2021) to identify the biological functions associated with the differentially expressed genes. The differentially expressed genes identified by the limma-voom analysis were used as input for GO enrichment analysis. Separate analyses were conducted for upregulated and downregulated genes between damaged and undamaged conditions in the early and late L3 stages. GO terms were considered significantly enriched if the adjusted *P*-value was less than or equal to 0.05.

### Transcription factor binding site analysis

The coordinates for regions classed as DR peaks ($LOG_2FC$ >0.5 and *P*adj <0.1) in either early or late L3 discs identified in (Harris et al., 2020) were converted from dm3 to the current dm6 version r6.59 using the coordinates converter tool on Flybase (flybase.org/convert/coordinates), and DNA sequences for each peak were extracted using the Extract Genomic DNA tool on Galaxy (usegalaxy.org) in FASTA format. The same process was used to obtain DNA sequences of static peaks, those that do not change between damaged and undamaged conditions ($LOG_2FC$ <0.5), which were used as the background sequences for TF enrichment analysis. The TF enrichment was performed using the AME tool on Meme Suite (meme-suite.org; McLeay and Bailey, 2010) with default options, assaying for motifs in the Fly Factor survey database. Identification of Stat92E and AP-1 binding sites in DR regions was performed using the FIMO tool on Meme Suite with default settings and a match *P*-value of *P*<0.001 (Grant et al., 2011).

### Acknowledgements

The authors thank Dr Iswar Hariharan and Dr Erika Bach for their generous gift of stocks and reagents. We thank the current members of the Harris lab for useful input and feedback. We thank the Bloomington *Drosophila* Stock Center and Developmental Studies Hybridoma Bank for stocks and reagents. We thank the ASU SOLS Biosciences Shared Resource Facility for use of equipment and expertise.

### Competing interests

The authors declare no competing or financial interests.

### Author contributions

Conceptualization: M.A.W., R.E.H.; Data curation: J.W.Q., M.C.L., R.E.H.; Formal analysis: J.W.Q., M.C.L., R.E.H.; Funding acquisition: R.E.H.; Investigation: J.W.Q., M.C.L., C.V.H.; Methodology: J.W.Q., M.C.L.; Project administration: M.A.W., R.E.H.; Resources: M.A.W., R.E.H.; Software: M.C.L., M.A.W.; Supervision: M.A.W., R.E.H.; Validation: J.W.Q., M.C.L., M.A.W.; Visualization: J.W.Q., M.C.L., M.A.W., R.E.H.; Writing – original draft: M.A.W., R.E.H.; Writing – review & editing: J.W.Q., M.C.L., C.V.H., M.A.W., R.E.H.

### Funding

 Deposited in PMC for immediate release.

### Data and resource availability

All relevant data can be found within the article and its supplementary information. Stocks are available upon request and details of stocks and reagents used in this study are available in the Materials and Methods. Raw sequencing data files are available on the Sequence Read Archive, BioProject Accession ID PRJNA1179511. The code for RNA-seq analysis is available at https://github.com/mariahlee/Regeneration-in-Drosophila-melanogaster.

### Peer review history

The peer review history is available online at https://journals.biologists.com/dev/lookup/doi/10.1242/dev.204632.reviewer-comments.pdf

### Special Issue

This article is part of the Special Issue 'Lifelong Development: the Maintenance, Regeneration and Plasticity of Tissues', edited by Meritxell Huch and Mansi Srivastava. See related articles at https://journals.biologists.com/dev/issue/152/20.

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
