## [Peer Review File · Development (Cambridge, England)]

A threshold level of JNK activates damage-responsive enhancers via JAK/STAT to promote tissue regeneration

John W. Quinn, Mariah C. Lee, Chloe Van Hazel, Melissa A. Wilson and Robin E. Harris
DOI: 10.1242/dev.204632

Editor: Kenneth Poss

Review timeline

Original submission:	7 January 2025
Editorial decision:	14 March 2025
First revision received:	18 June 2025
Accepted:	18 July 2025

Original submission

First decision letter

MS ID#: dev.204632

MS TITLE: A threshold level of JNK activates damage-responsive enhancers via JAK/STAT to promote tissue regeneration

AUTHORS: John Weston Quinn, Mariah C. Lee, Chloe Van Hazel, Melissa A. Wilson and Robin Harris

Dear Rob,

First of all I am very sorry about the delays, we had to chase a 3rd reviewer and ultimately decided to move forward with the 2 reviews. I have now received all the referees' reports on the above manuscript, and have reached a decision. The referees' comments are appended below, or you can access them online: please go to:

As you will see, the referees express considerable interest in your work, but have some significant criticisms and recommend a substantial revision of your manuscript before we can consider publication. In particular there are concerns about the conclusions that have been drawn from some of the presented data. If you are able to revise the manuscript along the lines suggested, which appears to involve further experiments, I will be happy receive a revised version of the manuscript. Your revised paper will be re-reviewed by one or more of the original referees, and acceptance of your manuscript will depend on your addressing satisfactorily the reviewers' major concerns. Please also note that Development will normally permit only one round of major revision. If it would be helpful, you are welcome to contact us to discuss your revision in greater detail. Please send us a point-by-point response indicating your plans for addressing the referees' comments, and we will look over this and provide further guidance.

Please attend to all of the reviewers' comments and ensure that you clearly highlight all changes made in the revised manuscript. Please avoid using 'Tracked changes' in Word files as these are lost in PDF conversion. I should be grateful if you would also provide a point-by-point response detailing how you have dealt with the points raised by the reviewers in the 'Response to Reviewers' box. If you do not agree with any of their criticisms or suggestions please explain clearly why this is so.

Reviewer 1

Advance summary and potential significance to field

This manuscript follows up on the authors' papers in 2016 and 2020, in which they discovered damage-responsive, maturity-silenced (DRMS) enhancers using *Drosophila* imaginal discs as a model. This manuscript focuses on a part of damage-responsive (DR) module. The authors utilized the DC ablation system, a recently developed method that induces acute, rather than persistent, injury, allowing for a mechanistic analysis of the cellular and molecular events involved in imaginal disc regeneration. Using *Drosophila* genetic tools and imaging techniques, the authors identified multiple interesting and important findings:

- 1) JNK signaling alone can initiate the regeneration program.
- 2) JNK signaling activity is spatially distinct.
- 3) Cooperation between JNK and JAK/STAT is crucial for DR enhancer activation in JNK^{low} cells.

Overall, experiments are well-conducted, and imaging data are of outstanding quality. This study addresses how multiple cis-regulatory elements and upstream signaling pathways coordinate to trigger enhancer activation upon injury. The central claim is significant, offering valuable insights into the complicated enhancer regulatory mechanisms governing tissue regeneration. However, I have several concerns as outlined below.

Comments for the author

Major comments

1. AP-1 activity and STAT signaling and/or Stat92E cis-regulatory elements in endogenous or full WNT DRMS enhancers

This manuscript primarily focuses on BRV-B or DR module, but the endogenous (or full) WNT enhancer contains additional activation (BRV-A) and MS modules. The authors mention the MS module in Fig 1, noting that it silences the WNT enhancer during regeneration in the late L3 stage. Experiments except this Fig. 1 investigate AP-1 and STAT activities within the context of the DR enhancer. How does AP-1 activity function in the context of DRMS or full WNT enhancer? To provide comprehensive analysis of full WNT enhancer activity, the authors require to examine whether AP-1 activities display distinct patterns (low, high like Fig 1 F,G and Fig. 7) when comparing with DRMS or full WNT enhancer (examining DRMS activity with AP-1 activity). Alternatively, the authors may examine whether Stat92E RNAi reduces DRMS or full WNT enhancer activity.

2. Quantification of AP-1 activity (Fig. 1F, G and Fig. 7) and lines 150-157 and 434-436

The authors describe "moderate", "weak", and "heterogenous levels" of JNK activity. However, there is no quantification data. Because at least 15 imaginal discs are used for each imaging analysis (Method), the authors could quantify JNK activity and DR enhancer activity by measuring their intensity for Dcp-1 negative cells (non-apoptotic cells) for Fig. 1F, G. For Fig. 7 (or Fig. 3E), the authors may quantify AP-1 intensity based on location, such as distal vs lateral pouch or center vs periphery.

Line 286 - 287 and Fig. 3G-H provide quantification data demonstrating dynamic fluorescence levels of AP-1 and STAT reporter genes across time points in the pouch. However, there is no data demonstrating AP-1 activity in central and peripheral regions.

3. N numbers

The authors indicate to use at least 15 discs in the Method section. This number is likely sufficient to make a conclusion. However, providing specific sample sizes in each figure legend would enhance transparency and clarity for readers.

4. Line 342 and Fig. S4

Although DR domains were used for Stat92E RNAi experiments of WNT and Mmp1 enhancers, DRMS was used for TDC1/2. Please change "DRMS WNT" and "DRMS Mmp1" to "DR WNT" and "DR Mmp1". Also, please state potential impact of MS for TDC1/2 DRMS activity. For example, the MS module's influence on the DRMS TDC1/2 enhancer may interfere with Stat92E RNAi effect.

Minor

1. Please clarify which developmental stages are used for experiments. It is challenging to know whether early L3, mid L3 or late L3 is used to examine AP-1, STAT, and DR enhancer activity. Please clarify in the text or method that the ablation is performed to examine enhancer activity for early, mid, or late L3 stage regeneration. Also, please clarify which stages the models in Fig. 7E-H apply to (e.g., early to mid L3 only or beyond).
2. In the text, Fig 6A and Fig 6B are flipped and referenced incorrectly.
3. Please provide details on how the Stat92E deletion construct was generated, including primer information and subcloning approaches, rather than stating that information is available upon request.
4. Fig. 1. What does "BP" mean? Please spell out for the first abbreviation. It could be base pair. If so, please provide a range rather than one number. The current could be starting base pair location.
5. "A.U." Please spell out for the first abbreviation.
6. Fig. 1. Stat92E(2n) - consensus sequence is TTCNGAA, which contain one n, not two n. Please clarify it. (May require putting one more N)
7. Fig 6A. Several aspects of the qPCR analysis in Fig. 6A need clarification: A) Please clarify what wild-type means. Does wild-type represent uninjured samples? Please clarify it. 2) Although the wild-type value would be one, this value should be shown in the graph. 3) Please provide individual value like other graphs, such as Fig. 6B. 4) Please clarify what A.U. indicates. This may be typo. Y-axis would be relative expression level. 5) Please perform statistical analysis and provide p-value.
8. Exact p-values should be reported for all graphs to improve data transparency.
9. Correct typo in Line 271 "a destabilized GFP reporter for JAK/STAT activity (10XSTAT-dGFP, (57),"

Reviewer 2

Advance summary and potential significance to field

This manuscript by Quinn et al. interrogates the mechanism of activation for a transcriptional enhancer during fly imaginal disc regeneration. While many regeneration enhancers have been identified, this story attempts to go one step further and uncover how the enhancer is activated. The proposed mechanism is that there is a class of regeneration enhancers that are activated by high JNK signaling alone, or by lower JNK signaling in combination with the activation of the JAK/STAT pathway. How regeneration enhancers are regulated is an important question making this model quite interesting. While the authors do support some parts, the data regarding JAK/STAT is insufficient to support their model. From the data presented, while the authors support the first part of their model, it appears at best that JAK/STAT co-regulates the DRWNT enhancer together with JNK more generally and this is less of a conceptual advance.

Comments for the author

Major concerns:

1. The authors have already published that JNK regulates the DRWNT enhancer, so it is not surprising that the DRWNT reporter overlaps with an AP-1 JNK controlled reporter. However, the DRWNT signal is so widespread that it is difficult to say how meaningful the colocalization is on its own. In addition, there appears to be plenty of GFP where there is no JNK reporter activity (and red where there is no green). The authors propose the attractive hypothesis that this may be from a low level of JNK signaling. They would have to do some quantitative analysis of their images to say this conclusively. That appears quite difficult to do with the breadth of the signal. Also, sometimes the DRWNT reporter has a widespread haze that would make quantification impossible

(Fig. 1J, Fig. 4A, Fig. 6C, and Fig. 7CD). The authors need to better establish that there are actually two populations of cells.

2. In Fig. 3A-D it is very difficult to detect upregulation of the 10XSTAT reporter with the level of background fluorescence. The quantification in Fig. 3G shows at most a 20% increase. The images in Fig. 3M and N are better, yet it remains difficult to observe the extent that the 10XSTAT reporter colocalizes with the DRWNT reporter. Colocalization is also difficult to observe between the JNK "marker" Mmp1 in Fig 3M and N. While there is some overlap, the widespread expression of the reporter makes it difficult to tell if it is meaningful and there is plenty of signal that does not colocalize.

3. The premise of this manuscript is that JAK-STAT regulates the DRWNT enhancer in some conditions. Besides colocalization which is both correlative and difficult to interpret (see above), the two pieces of evidence that support that claim are insufficient (the DRWNT reporter with STAT92E knockdown, the DRWNT reporter with STAT binding sites deleted).

First, the entire DRWNT reporter appears less bright with STAT92E knockdown. The reduction in signal is not confined to the periphery but also occurs where JNK signaling is highest. Likely, JAK-STAT signaling is just centrally required for full activation of DRWNT. Furthermore, images in Fig. 4A are not representative of the quantification shown in Fig. 4B. It appears that the DRWNT reporter is less bright at every time point but the quantification shows that it is almost identical at some (see 6h and 24h time points). Also, does the GFP only go up 2X from 0h to 18h in the control? Something does not make sense with this quantification as it is currently presented. Finally, why not test expression regulated by the endogenous enhancer rather than solely rely on a reporter?

Second, do we know if insertional effects are leading to the differences observed between the DRWNT reporter and the reporter with $\Delta_{STAT-DR}$? One would have to quantify many different transgenic insertions to make a statement that the differences are specific to the sequence changes.

Minor comments:

1. The authors use a minimal hsp70 promoter in their enhancer promoter constructs. Does this minimal promoter activate on its own without an enhancer? Forgive us if this is already published elsewhere but please at least provide a reference.
2. line 149 - do imaginal discs form a blastema? Even in an ablation model?
3. line 180 - the experiments show that DRWNT activates without apoptosis, but it did not test the dependence on JNK. The language is too strong.
4. line 188-192 - This statement belongs in the Discussion and not the Results section. Far too speculative.
5. line 192-195 - This is an over-statement. The data only shows two things. First, that a JNK reporter and DRWNT are correlated by colocalization. Second, that DRWNT does not depend on apoptosis eliminating the requirement of the feed forward mechanism.
6. line 358-359 - This is an over-statement. The authors show in these experiments that Stat92E is important for promoting regeneration, but the authors do not show with these experiments that it has anything to do with DRWNT.

First revision

Author response to reviewers' comments

We greatly appreciate both reviewers taking the time to provide comments and feedback on our work. The suggestions were insightful, reasonable and helpful, and the authors agree that the suggested changes will only act to strengthen the paper. As such we have done our best to address each of these suggestions, and in addition we have reworked the descriptions of these data in the result section to provide clarity and reduce its length.

Reviewer 1 major comments

1. AP-1 activity and STAT signaling and/or Stat92E cis-regulatory elements in endogenous or full WNT DRMS enhancers

This manuscript primarily focuses on BRV-B or DR module, but the endogenous (or full) WNT enhancer contains additional activation (BRV-A) and MS modules. The authors mention the MS module in Fig 1, noting that it silences the WNT enhancer during regeneration in the late L3 stage. Experiments except this Fig. 1 investigate AP-1 and STAT activities within the context of the DR enhancer. How does AP-1 activity function in the context of DRMS or full WNT enhancer? To provide comprehensive analysis of full WNT enhancer activity, the authors require to examine whether AP-1 activities display distinct patterns (low, high like Fig 1 F,G and Fig. 7) when comparing with DRMS or full WNT enhancer (examining DRMS activity with AP-1 activity). Alternatively, the authors may examine whether Stat92E RNAi reduces DRMS or full WNT enhancer activity.

This work primarily focuses on the activation of the enhancer, which we previously showed is achieved via the DR^{WNT} region (Harris et al 2016). Thus, we focused on this enhancer region to understand the major regulatory inputs without the inhibitory module. However, it is certainly possible that the MS region of the full enhancer could influence its behavior, and so we examined the behavior of the full DRMS in terms of activation level by AP-1 (S1F Fig and lines 152-157) and the effect of Stat92E^{RNAi} on the full DRMS (S4A-B Fig, quantified in S4E Fig, and lines 311-315), as suggested by the reviewer. These findings show that the behavior of the full DRMS region is similar to that of the DR region in both cases.

2. Quantification of AP-1 activity (Fig. 1F, G and Fig. 7) and lines 150-157 and 434-436

The authors describe "moderate", "weak", and "heterogenous levels" of JNK activity. However, there is no quantification data. Because at least 15 imaginal discs are used for each imaging analysis (Method), the authors could quantify JNK activity and DR enhancer activity by measuring their intensity for Dcp-1 negative cells (non-apoptotic cells) for Fig. 1F, G. For Fig. 7 (or Fig. 3E), the authors may quantify AP-1 intensity based on location, such as distal vs lateral pouch or center vs periphery.

Line 286 - 287 and Fig. 3G-H provide quantification data demonstrating dynamic fluorescence levels of AP-1 and STAT reporter genes across time points in the pouch. However, there is no data demonstrating AP-1 activity in central and peripheral regions.

We have added quantification of the results as suggested by the reviewer, evaluating the proportion of cells in ablated discs with JNK activity and with DR enhancer activity, each with and without Dcp-1 activity (S1A-C and lines 138-142).

To strengthen the findings illustrated in Fig 7 and Fig3E, which demonstrate the heterogenous levels of JNK activity across a wound, we have now added in quantification of this activity by generating quantified profiles of AP-1 fluorescence across a wound (Fig 1G and lines 148-150), which clearly demonstrates the existence of different JNK levels in the center and periphery of the injury.

3. N numbers

The authors indicate to use at least 15 discs in the Method section. This number is likely sufficient to make a conclusion. However, providing specific sample sizes in each figure legend would enhance transparency and clarity for readers.

We have added the sample sizes to all the figure legends where appropriate to clarify these data.

4. Line 342 and Fig. S4

Although DR domains were used for Stat92E RNAi experiments of WNT and Mmp1 enhancers, DRMS was used for TDC1/2. Please change "DRMS WNT" and "DRMS Mmp1" to "DR WNT" and "DR Mmp1". Also, please state potential impact of MS for TDC1/2 DRMS activity. For example, the MS module's influence on the DRMS TDC1/2 enhancer may interfere with Stat92E RNAi effect.

We have altered the DRMS labels as suggested (line 332).

When considering the impact of the MS for TDC1/2, it is important to note that this DRMS enhancer at the TDC1/2 gene is not as well characterized as the enhancers at the WNT and Mmp1 loci. For example, the DR and MS regions have not been separately identified, as they have for the others. As such, it is difficult to evaluate potential input from the MS separate from the DR. We have added a line in the text to indicate this information (line 329). However, since we have examined the potential influence of the MS region for the enhancer at the WNT locus (Major comment 1 from this reviewer), and found no obvious contribution, these findings may also help to address this concern.

Minor

1. Please clarify which developmental stages are used for experiments. It is challenging to know whether early L3, mid L3 or late L3 is used to examine AP-1, STAT, and DR enhancer activity. Please clarify in the text or method that the ablation is performed to examine enhancer activity for early, mid, or late L3 stage regeneration. Also, please clarify which stages the models in Fig. 7E-H apply to (e.g., early to mid L3 only or beyond).

Information about developmental stages has been added to the methods text, indicating that all discs in the paper are of the early L3 stage, (except for panel Figure 1B) and imaged at the hours after heat shock indicated in each panel. This information has also been added to the model figure legend (Figure 7) as suggested for clarity.

2. In the text, Fig 6A and Fig 6B are flipped and referenced incorrectly.

The figure/text has been corrected.

3. Please provide details on how the Stat92E deletion construct was generated, including primer information and subcloning approaches, rather than stating that information is available upon request.

Cloning details and primer sequence information has been added to the materials and methods (lines 607-619).

4. Fig. 1. What does "BP" mean? Please spell out for the first abbreviation. It could be base pair. If so, please provide a range rather than one number. The current could be starting base pair location.

All figures showing the schematics in question (Fig 1A, S54G and S4H) have been altered, adding the genome coordinates of the enhancers, and the base pair ranges for the TF sites indicated within these regions.

5. "A.U." Please spell out for the first abbreviation.

Text has been changed throughout to address this comment.

6. Fig. 1. Stat92E(2n) - consensus sequence is TTCNGAA, which contain one n, not two n. Please clarify it. (May require putting one more N)

This typo has been corrected.

7. Fig 6A. Several aspects of the qPCR analysis in Fig. 6A need clarification: A) Please clarify what wild-type means. Does wild-type represent uninjured samples? Please clarify it. 2) Although the wild-type value would be one, this value should be shown in the graph. 3) Please provide individual value like other graphs, such as Fig. 6B. 4) Please clarify what A.U. indicates. This may be typo. Y-axis would be relative expression level. 5) Please perform statistical analysis and provide p-value.

The information regarding the qPCR analysis has been modified to address these concerns: 1) wild type has been clarified in the figure legend, 2) the wild type value has been added to both Fig 6A and B, 3) the individual data points have been added to the graph, 4) A.U. was a typo that has now been corrected to fold change (F.C.), 5) statistical analyses have been performed, and indicated in the figure legend with a p value.

8. Exact p-values should be reported for all graphs to improve data transparency.

P values have been reported for all graphs as requested.

9. Correct typo in Line 271 "a destabilized GFP reporter for JAK/STAT activity (10XSTAT-dGFP, (57),"

We weren't clear on the typo being referred to, but have changed the wording of this sentence (line 265) and changed the nomenclature to "10XSTAT-DGFP" to reflect the original use in the cited paper. We will happily make further corrections if necessary.

Reviewer 2 major comments:

1. The authors have already published that JNK regulates the DRWNT enhancer, so it is not surprising that the DRWNT reporter overlaps with an AP-1 JNK controlled reporter. However, the DRWNT signal is so widespread that it is difficult to say how meaningful the colocalization is on its own. In addition, there appears to be plenty of GFP where there is no JNK reporter activity (and red where there is no green). The authors propose the attractive hypothesis that this may be from a low level of JNK signaling. They would have to do some quantitative analysis of their images to say this conclusively. That appears quite difficult to do with the breadth of the signal. Also, sometimes the DRWNT reporter has a widespread haze that would make quantification impossible (Fig. 1J, Fig. 4A, Fig. 6C, and Fig. 7CD). The authors need to better establish that there are actually two populations of cells.

To address this concern and ensure that we clearly show that the DR^{WNT} expressing cells have both high JNK and low JNK levels, we have taken advantage of a recently generated transgenic fly we created for this purpose, which expresses a nuclear localized fluorescent reporter under the control of the enhancer, DR^{WNT}(NLS), clearing up the somewhat hazy signal of the original reporter, which is cytosolic. We have used this new reporter alongside high magnification imaging (Fig1F-F''''', lines 146-148) to show clearly the different levels of AP-1 that exist in the disc (quantified in Fig 1G) and how they overlap with the DRWNT enhancer activity. We believe these data greatly strengthen our described observations of the different JNK levels and coincidence with the enhancer. We have retained the original images using the cytosolic reporter, quantified their signal and overlap with Dcp-1 (reviewer 1, comment 2) and added them to the supplemental data (S1A-C Fig).

2. In Fig. 3A-D it is very difficult to detect upregulation of the 10XSTAT reporter with the level of background fluorescence. The quantification in Fig. 3G shows at most a 20% increase. The images in Fig. 3M and N are better, yet it remains difficult to observe the extent that the 10XSTAT reporter colocalizes with the DRWNT reporter. Colocalization is also difficult to observe between the JNK "marker" Mmp1 in Fig 3M and N. While there is some overlap, the widespread expression of the reporter makes it difficult to tell if it is meaningful and there is plenty of signal that does not colocalize.

We agree that the background signal of the 10XSTAT reporter can unfortunately sometimes hinder imaging in these experiments. As such, we have repeated several experiments using a different staining protocol with anti-GFP staining and changed the co-stained factors, substituting the original images in Fig 3A-C and Fig3 K-M. We have also re-analyzed images bearing the STAT reporter to improve the overall image quality (S3A Fig), which together better illustrates the expression distribution and overlaps with other factors in these experiments.

3. The premise of this manuscript is that JAK-STAT regulates the DRWNT enhancer in some conditions. Besides colocalization which is both correlative and difficult to interpret (see above), the two pieces of evidence that support that claim are insufficient (the DRWNT reporter with STAT92E knockdown, the DRWNT reporter with STAT binding sites deleted).

First, the entire DRWNT reporter appears less bright with STAT92E knockdown. The reduction in signal is not confined to the periphery but also occurs where JNK signaling is highest. Likely, JAK-STAT signaling is just centrally required for full activation of DRWNT. Furthermore, images in Fig. 4A are not representative of the quantification shown in Fig. 4B. It appears that the DRWNT reporter is less bright at every time point but the quantification shows that it is almost identical at some (see 6h and 24h time points). Also, does the GFP only go up 2X from 0h to 18h in the control? Something does not make sense with this quantification as it is currently presented. Finally, why not test expression regulated by the endogenous enhancer rather than solely rely on a reporter?

We appreciate this comment and understand the reviewer's concern, since this evidence is key to the proposed role for JAK/STAT signaling in the regulation of DR^{WNT} enhancer activity. As such, we have addressed the issue in several ways by both improving the existing data and adding new experimental data.

Firstly, we agree that the images of the DR^{WNT} reporter in *Stat92E* knockdown that were presented appeared to be overall dimmer. We have included images we consider are more representative of

the phenotype we observe (Fig 4A') show the lack of peripheral enhancer activity, and presented them without the DAPI channel to show the GFP levels more clearly. This was also re-quantified (Fig 4B), which now shows an even more obvious difference between enhancer activity in the two conditions.

We did the same for the DR^{WNT} deltaSTAT enhancer reporter (Fig 4F), which has an overall smaller footprint than the DR^{WNT} enhancer reporter. This is mainly due to the lack of DR^{WNT} in the periphery, however it is possible that any reduction of activity in the center of the wound could be due to JAK/STAT also having a minor role in the initial activation of DR^{WNT} enhancer activity early in wound formation, when JNK and JAK/STAT levels are both high in the center of the wound. This possibility does not alter the conclusion that the primary role for JAK/STAT is to ensure the spread of DR^{WNT} activity into the wound periphery. To further confirm this idea, and strengthen our model, we performed an additional experiment to limit JAK/STAT signaling in only one half of the pouch, using *hh-GAL4;UAS-Stat92E^{RNAi}* (Fig 4E, lines 320-324). This reduces the peripheral DR^{WNT} activity but doesn't strongly influence the activity in the center, showing that JAK/STAT has an important function promoting DR^{WNT} activity beyond the initial boundary of the injury.

Finally, regarding the use of the endogenous enhancer, we understand this to mean using Wg signaling as a readout. Unfortunately, it is clear from previous work that Wg has numerous regulatory inputs in both development and regeneration that complicate its use as a single readout, and therefore we have limited our analysis to the isolated DR^{WNT} (and DRMS^{WNT}) enhancer region(s).

Second, do we know if insertional effects are leading to the differences observed between the DRWNT reporter and the reporter with ΔSTAT-DR? One would have to quantify many different transgenic insertions to make a statement that the differences are specific to the sequence changes.

The transgenes in question, DR^{WNT} and DR^{WNT} deltaSTAT (and others that are directly compared, for example the DRMS^{WNT} and DR^{WNT}) are in the same genomic landing site, and therefore have no insertional effects. We will have included this information in the material and methods (line 616).

Minor comments:

1. The authors use a minimal hsp70 promoter in their enhancer promoter constructs. Does this minimal promoter activate on its own without an enhancer? Forgive us if this is already published elsewhere but please at least provide a reference.

The reference related to this comment and information on the hsp70 promoter has been added to the materials and methods (line 614-616).

2. line 149 - do imaginal discs form a blastema? Even in an ablation model?

The use of blastema is generally an accepted terminology in this model, as defined by numerous publications that use this term to refer to cells in *Drosophila* imaginal discs that lose their differentiated (or determined) identity, gain plasticity, increase proliferation, occur locally around a wound and contribute to the regenerated tissue following physical injury, irradiation or genetic ablation (see for examples Abbott et al., 1981; Kiehle and Schubiger, 1985; Bosch et al., 2005; Bergantinos et al., 2010, Smith-Bolton, 2016, Fox et al. 2020, Worley and Hariharan, 2022).

3. line 180 - the experiments show that DRWNT activates without apoptosis, but it did not test the dependence on JNK. The language is too strong.

The language has been changed to address this comment. (line 179).

4. line 188-192 - This statement belongs in the Discussion and not the Results section. Far too speculative.

This statement has been removed (line 189).

5. line 192-195 - This is an over-statement. The data only shows two things. First, that a JNK reporter and DRWNT are correlated by colocalization. Second, that DRWNT does not depend on apoptosis eliminating the requirement of the feed forward mechanism.

This statement has been removed (line 189).

6. line 358-359 - This is an over-statement. The authors show in these experiments that Stat92E is important for promoting regeneration, but the authors do not show with these experiments that it has anything to do with DRWNT.

The language has been changed to address this comment. (line 348-351).

Second decision letter

MS ID#: dev.204632R1

MS TITLE: A threshold level of JNK activates damage-responsive enhancers via JAK/STAT to promote tissue regeneration

AUTHORS: John Weston Quinn, Mariah C. Lee, Chloe Van Hazel, Melissa A. Wilson and Robin Harris

Dear Rob,

I am happy to tell you that your manuscript has been accepted for publication in Development, pending our standard publication integrity checks.

Reviewer 1

Advance summary and potential significance to field

The authors address my comments. I have only one minor comment.

Comments for the author

Minor

Fig. 1G and Quantification Method: The X-axis label ("distance (% distal pouch)") is not clearly defined. Providing a more detailed explanation of how this axis is calculated - either in the Methods section or the figure legend - would strengthen the presentation of the data. For example, clarify whether the center of the wound area is designated as 50% and how much line is extended.

Reviewer 2

Advance summary and potential significance to field

The authors addressed all of the concerns that were presented in the first review.